# Comparison of Cassava Chips and Winged Bean Tubers with Various Starch Modifications on Chemical Composition, the Kinetics of Gas, Ruminal Degradation, and Ruminal Fermentation Characteristics Using an In Situ Nylon Bag and an In Vitro Gas Production Technique

**DOI:** 10.3390/ani13101640

**Published:** 2023-05-15

**Authors:** Narirat Unnawong, Chaichana Suriyapha, Benjamad Khonkhaeng, Sompong Chankaew, Teppratan Rakvong, Sineenart Polyorach, Anusorn Cherdthong

**Affiliations:** 1Department of Animal Science, Tropical Feed Resources Research and Development Center (TROFREC), Faculty of Agriculture, Khon Kaen University, Khon Kaen 40002, Thailand; nariratunnawong@kkumail.com (N.U.); chaichana_s@kkumail.com (C.S.); 2Department of Animal Science, Faculty of Agricultural Innovation and Technology, Rajamangala University of Technology Isan, Nakhon Ratchasima Campus, Nakhon Ratchasima 30000, Thailand; kbenjamad211223@gmail.com; 3Department of Agronomy, Faculty of Agriculture, Khon Kaen University, Khon Kaen 40002, Thailand; somchan@kku.ac.th (S.C.); teppratan_r@kkumail.com (T.R.); 4Department of Animal Production Technology and Fisheries, Faculty of Agricultural Technology, King Mongkut’s Institute of Technology Ladkrabang, Bangkok 10520, Thailand; sineenart.po@kmitl.ac.th

**Keywords:** feed energy source, by-pass starch, resistant starch, winged bean, ruminant

## Abstract

**Simple Summary:**

Cassava chips (CSC) have typically been used as an energy source for ruminant rations in tropical regions due to their high starch content and higher degradability rate in the rumen, which is higher than 90%. However, inconsistencies in production and fluctuations in prices encourage animal nutritionists to search for alternatives to CSC. The introduction of new tuberous plant species may help improve local feed diversity while also reducing feed shortages in particular areas. The winged bean tuber (WBT) has a high proportion of starch that might be able to replace CSC as an energy source. Moreover, it was discovered that steamed WBT might be useful for improving feed efficiency, which could lower rapid starch degradability and maintain rumen pH when compared to the WBT raw material.

**Abstract:**

This research assessed the impact of cassava chips (CSC) and winged bean tubers (WBT) with various starch modification methods on the chemical composition, ruminal degradation, gas production, in vitro degradability, and ruminal fermentation of feed using an in situ and in vitro gas production technique. Experimental treatments were arranged for a 2 × 5 factorial, a completely randomized design with two sources of starch and five levels of modification treatments. Two sources of starch were CSC and WBT, while five modification treatments of starch were: no modification treatment, steam treatment, sodium hydroxide (NaOH) treatment, calcium hydroxide (CaOH2) treatment, and lactic acid (LA) treatment. The starch modification methods with NaOH and CaOH_2_ increased the ash content (*p* < 0.05), whereas the crude protein (CP) content was lower after treatment with NaOH (*p* < 0.05). Steam reduced the soluble fraction (a) and effective dry matter degradability of WBT in situ (*p* < 0.05). In addition, the WBT steaming methods result in a lower degradation rate constant in situ (*p* < 0.05). The degradation rate constants for the insoluble fraction (c) in the untreated CSC were higher than those of the other groups. Starch modification with LA reduced in vitro dry matter degradability at 12 and 24 h of incubation (*p* < 0.05). The starch modification method of the raw material showed the lowest pH value at 4 h (*p* < 0.05). The source of starch and starch modification methods did not influence the in vitro ammonia nitrogen concentrations, or in vitro volatile fatty acids. In conclusion, compared to the CSC group and untreated treatment, treating WBT with steam might be a more effective strategy for enhancing feed efficiency by decreasing or retarding ruminal starch degradability and maintaining ruminal pH.

## 1. Introduction

Cassava (*Manihot esculenta*) is a commonly grown tuberous tropical field crop, especially in the northeast of Thailand [1,2,3]. Cassava roots can be chopped, processed, and dried into cassava chips (CSC) for use as an energy source in feed. They have a high soluble carbohydrate content (75–85%) but a low crude protein (CP) content (2–3%) [2]. Cassava starch has a high solubility and an immediate degradability rate in the rumen of more than 90% [4]. Although cassava is widely used as a chewable feed in the tropics, there are still some problems with productivity variations due to the effects of climate change on agricultural production, which also lead to instability in prices [3]. The incorporation of new tuberous species of plants or other plants may help to improve local feed diversity while also reducing feed limitations in certain locations [5].

It has been recommended that *Psophocarpus tetragonolobus* (winged bean; WB) be used more widely, particularly in tropical areas [6]. The WB has green pod productivity of up to 10 t/ha and tuber productivity of up to 11 t/ha [5]. The WB tuber (WBT) has 20% CP, 3.0% crude fiber (CF), 0.6% fat, and a high starch level of 25–30% (energy = 3.82 Mcal/kg) [6,7], which should make it a suitable alternative for animal feed [3].

Usually, high proportions of carbohydrate sources are often used for high-producing ruminants to support high milk production and quick weight gain [8]. However, the rapid fermentation of high-carbohydrate feeds containing high starch causes a drop in ruminal pH and increases the risk of rumen acidosis [8]. Therefore, more research has been done to find a suitable way of modulating the degradability of starch sources in the rumen and boosting feed efficiency by partially moving starch digestion to the small intestine [9,10]. By increasing resistant starch in the rumen, feed processing techniques, including physical and chemical approaches, can alter starch, reduce rumen-degradable starch, and improve starch flow to the duodenum. It was observed that steaming treatment may effectively control ruminal degradability and improve by-pass starch [10,11].

Steaming is a physical method for modifying starch that improves starch digestion in the rumen [12,13]. In addition, the rate of starch breakdown in ruminant feed is often decreased by treatment with less expensive chemicals, such as alkalis, including sodium hydroxide (NaOH) and calcium hydroxide (CaOH_2_), as well as lactic acid (LA) [4,14].

Although BWT contains a lot of starch, it can rapidly break down in the rumen, resulting in inadequate utilization. The novelty of the current work is due to the lack of knowledge used to control starch digestion in the rumen for BWT and the appropriate tests for in vitro investigation. This study tested the hypothesis that a modified starch product obtained from WBT could be used as a potential energy source, decrease the rapid ruminal degradation rate, and maintain ruminal pH. The aim was to evaluate the chemical composition, ruminal degradation, gas kinetics, and ruminal fermentation of CSC and WBT with various modified starches using an in situ degradability and in vitro gas production technique.

## 2. Materials and Methods

All of the methodology and procedures involved in this research were approved by the Ethical Committee under the Institutional Guidelines of Khon Kaen University, National Research Council of Thailand (record no. IACUC-KKU-46/65) to ensure animal welfare.

### 2.1. Experimental Station and Treatment Preparation

The cassava chips (CSC) were obtained from Khon Kaen University’s animal feed factory, Department of Animal Science, Faculty of Agriculture, and WBT was provided by the Department of Agronomy, Faculty of Agriculture, Khon Kaen University, Thailand (16°26′48.16” N, 102°49′58.8” E). The steam treatment method followed the procedure of Srakaew et al. [8]. The CSC and WBT were soaked in water for 12 h to increase moisture content before steaming at 100 °C for 45 min in a 2-tiered steam pot (20 cm high × 80 cm^2^ in diameter) and sun-drying for 48 h until the moisture content was less than 10%. The NaOH treatment method followed the process of Srakaew et al. [8]. The CSC and WBT were treated with a 3.5% NaOH solution at a ratio of 1:1 (vol/wt) for 24 h, followed by sun drying for 48 h. The CaOH_2_ treatment method was taken from the procedure of Wanapat et al. [15]. CSC and WBT were treated with a 2.0% CaOH_2_ solution at a 1:1 (vol/wt) ratio, followed by 48 h of anaerobic fermentation in bottles and then 48 h of sun drying.

The LA treatment method was performed using the method of Pilachai et al. [4] with 1.0% LA at a ratio of 1:1 (vol/wt). CSC and WBT were steeped in LA solution and left uncovered for 48 h. Then, for 48 h, sun drying was done to remove moisture. Finally, all of the treated CSC and WBT were ground to the same 2–3 mm particle size and passed through a 2–4 mm sieve.

### 2.2. Experimental Design and Dietary Treatments

The present study was performed with different incubation intervals and two sub-experiments using a nylon bag measurement and gas production technique. In both sub-experiments, a 2 × 5 factorial experiment design was carried out and set up using a completely randomized design (CRD) with three replication runs. The diets used in the experiments had two starch energy sources, CSC and WBT, with five modified starch treatment methods: untreated, steam-treated, NaOH-treated, CaOH_2_-treated, and LA-treated.

To prepare the experimental diet for the chemical composition test, the nylon bag test, and the gas production test, all of the dietary samples were oven-dried for 48 h at 60 °C, and ground and passed through a 1-mm sieve (Cyclotech Mill, Tecator, Sweden). Experimental dietary samples were analyzed for their chemical compositions with the standard method of the AOAC [16], including their dry matter (DM, no. 967.03), ash (no. 492.05), CP (no. 984.13), and ether extract (EE, no. 920.39) content. The neutral detergent fiber (NDF) and acid detergent fiber (ADF) contents of the samples were analyzed according to the procedure of Van Soest et al. [17]. Non-fiber carbohydrate (NFC) was calculated according to the procedure of Holtshausen [18]:NFC (%) = 100 − (CP (%) + EE (%) + NDF (%) + ash (%))(1)

Using an adiabatic calorimeter bomb, gross energy (GE) was calculated (AC500, LECO Corporation, St. Joseph, MI, USA).

### 2.3. In Situ Nylon Bag Measurement

Three male ruminally fistulated Thai native beef cattle (White Lamphun) with body weights (BW) of 325 ± 25.0 kg were used in this experiment. All cattle received a concentrate diet of 1.0% BW (16.0% CP and 10.46 MJ of ME/kg DM) and rice straw twice daily ad libitum. All animals were housed in individual 2 × 5 m^2^ cages within an evaporative barn and freely watered with individual nipple cups. All animals were fed the diet for the adaptation period of 14 days prior to the experiment.

The in situ nylon bag approach was used to determine DM disappearance, according to Ørskov and McDonald [19]. After 48 h of oven drying at 60 °C, all samples were crushed and passed through a 2-mm screen. For all modified and non-modified samples, approximately 4 g of feed sample was collected in pre-weighed nylon bags (12 × 6 cm^2^, 40 m average pore size). All of the nylon bags were fastened to an iron chain with nylon threads and then left hanging in the rumens of three fistulated cattle for 2, 4, 8, 12, 16, 24, and 48 h. Using triplicates based on each incubation time point, the samples of each treatment were incubated concurrently in three animals.

All sample test bags were taken out of the rumen after each incubation period and handwashed in lukewarm water for 10 min or until the wash water was clear. The following step involved drying nylon bags holding mixed test feed leftovers in an air oven for 48 h at 60 °C. We gathered all bag feed samples for their appropriate blanks, and a blank bag containing no sample for each removal time. The samples at 0 h of incubation underwent a similar procedure for washing and drying in an oven. Then, the difference in weight before and after incubation was used to calculate the DM loss.

Data for DM ruminal disappearance features were fitted to an exponential equation using the NEWAY program, following the method outlined by Ørskov and McDonald [19]. The disappearance rate at time *t* (%) *p* was calculated as follows:*p* = a + b (1 − e^−ct^)(2)
where a is the intercept of the degradation curve at time zero (%), b is the fraction of DM that degraded in the rumen (%), c is the rate constant of the disappearance of fraction b (h^−1^), and t is the time of incubation (h).

The following equation was used to determine the effective degradability (ED) of dry matter (EDDM) and the effective degradability of organic matter (EDOM). Drying at 100 °C for 24 h yielded the DM, and ashing at 550 °C for 4 h yielded the OM.
ED = a + [bc/(c + k)](3)
where: k is the assumed rate of particulate outflow from the rumen (0.05 h^−1^) [19].

### 2.4. Ruminal Liquor Preparation and In Vitro Gas Technique

In order to obtain a representative rumen fluid, four native Thai beef cows were considered. Four native Thai beef cows weighing 350 ± 20 kg each were employed as donors for ruminal liquor. To obtain ruminal liquor, the animals were fed a concentrate diet ad libitum at 0.5% of BW daily (14.0% CP and 80.5% total digestible nutrients; TDN), along with roughage (rice straw) (6:30 a.m. and 4:30 p.m.). Individual enclosures were employed to house the cattle, and free access to mineral blocks and clean water was provided. For 21 days, the cattle were fed a diet before obtaining rumen liquor. Before morning feeding, ruminal liquor from each animal was collected using a stomach tube connected to a vacuum pump. The ruminal liquor was passed through five layers of cheesecloth before being placed in pre-heated thermos bottles at 39 °C and transported to the laboratory.

A sample containing 0.5 g of substrate (CSC, WBT, treated CSC, and treated WBT) was weighed out and put into 40 mL bottles. The rumen fluid was taken to the laboratory and combined with freshly made artificial saliva at a 2:1 ratio to form an inoculum. The process described by Sommart et al. [20] was used to create artificial saliva. In an anaerobic environment, rumen medium preparations, including distilled water (2190 mL), a trace mineral mixture (0.46 mL), a macro mineral mixture (730 mL), a resazurine mixture (2 mL), a reduction mixture (120 mL), and a buffer mixture (1460 mL), were combined with rumen liquor (1210 mL). The ground WBT samples were weighed (at 500 mg) in 50 mL bottles at their respective levels of total substrate. A 40 mL volume of ruminal liquor medium that had been warmed up was injected by hand into substrate-containing bottles. Three replication runs, three bottles per treatment (10 treatments + 3 bottles of blank), and the measurement of the gas kinetics and cumulative gas output were used. At 0, 0.5, 1, 2, 4, 6, 8, 12, 18, 24, 48, 72, and 96 h of incubation, the amount of gas generated was measured. The amount of gas produced was measured using a 20 mL glass aloe hypodermic syringe (U4520, Becton, Dickinson and Company, Franklin Lakes, NJ, USA). An 18-gauge injection needle was used to pierce the bottles inside the heating chamber. The 60 bottles were prepared independently for pH, NH_3_-N, and volatile fatty acid (VFA) analyses. For the NH_3_-N and VFAs analyses, a total of 10 mL of liquid samples were employed. Another set of 60 bottles was used for a digestibility analysis (3 bottles per treatment × 10 treatments × 2 sample times: 12 and 24 h incubation).

The first set was used for the evaluation of gas production. A time series of incubations was conducted at 0, 1, 2, 4, 6, 8, 10, 12, 18, 24, 48, 72, and 96 h. The observed gas data were then entered into the following equation [19]:Y = a + b (1 − e^(−ct)^)(4)
where: Y is the production of gas at time *t* (mL), a is the immediate production of gas from the soluble fraction (mL), b is the production of gas from the insoluble fraction (mL), c is the gas production rate constant for the insoluble fraction (mL h^−1^), t is the incubation time (h), and a + b = is the potential extent of gas production (ml).

Using the second set, and a digital pH meter (HANNA Instrument (HI) 8424 microcomputer, Singapore), the ruminal pH was measured after 4 h of incubation. The 20 mL was maintained in 5 mL of 1 M H_2_SO_4_ and held at 20 °C for ammonia nitrogen (NH_3_-N) analysis using a spectrophotometer (UV/VIS spectrophotometer, PG Instruments LtD., London, United Kingdom) based on the method by Fawcett and Scott [21]. Then, in vitro volatile fatty acid (VFA) concentration analysis was carried out in accordance with Porter and Murray [22]. This was done using a gas chromatograph (Newis GC-2030: Shimadzu, Shimadzu Corporation, Kyoto, Japan) outfitted with a capillary column (DB-Wax column: 30-m length, 0.25 mm diameter, 0.25 m film; Agilent Technology, Santa Clara, CA, USA) according to the previous study [23].

The in vitro digestibility of dry matter (IVDMD) was evaluated after 12 and 24 h of incubation when the contents were filtered through pre-weighed Gooch crucibles, and residual DM was quantified. The percent weight loss was calculated and displayed as IVDMD. The IVDMD (%) was computed as follows:IVDMD = (((RS100 − C) − (RB100 − C))/WS) × 100,(5)
where RS100 is the weight of the crucible and the residue after 100 °C drying, RB100 is the weight of the crucible and the chemical reagent residue after 100 °C drying (blank), C is the weight of the dried crucible, and WS is the weight of the sample (before incubation) on DM. The dried feed sample and residue left above were ashed for 6 h at 550 °C to determine in vitro organic matter digestibility (IVOMD) [24].

### 2.5. Statistical Analysis

All data were subjected to an analysis of variance (ANOVA), which was carried out using SAS PROC GLM [25]. The data were analyzed using the following model:Yijk = µ + Ai + Bj + Abij + Ɛijk(6)
where: Yijk is the observations; µ is the overall mean; Ai is the effect of factor A (CSC and WBT), Bj is the effects of factor B (untreated, steam treated, NaOH treated, CaOH_2_ treated, and LA treated), Abij is the interaction between factor A and B, and Ɛijk is the residual effect. Tukey’s multiple comparison test was used to determine whether there were significant differences between treatments [26], and statistically significant differences were accepted at *p* ≤ 0.05.

## 3. Results

### 3.1. Chemical Compositions of Cassava Chip- and Winged Bean Tuber-Modified Starch

The chemical composition comparison of CSC- and WBT-modified starch is presented in Table 1. All parameters had no interaction effects between the starch source and the starch modification procedures. There was no change in the DM of any of the dietary samples. The sources of starch in the CP, EE, and ADF of the WBT group were higher (*p* < 0.05) than those of the CSC group (185.18, 11.97, and 61.89 g/kg DM, respectively), while the NDF and NFC contents were lower (*p* < 0.05). Nevertheless, the source of starch had no effect on the ash or GE content.

As a result of the starch modification methods, the ash content increased (*p* < 0.05) when the starch was treated with NaOH and CaOH_2_. However, the CP content decreased after treatment with NaOH, and there was also a lower NFC content in the CaOH_2_ group (*p* < 0.05). Furthermore, after steam treatments, GE in CSC and WBT increased (*p* < 0.05).

### 3.2. In Situ Degradability

The effects of rumen incubation on the kinetics of degradation of the DM and OM characteristics of CSC- and WBT-modified starch are presented in Table 2. The results show an interaction effect between the starch source and the modified procedures on the soluble fraction (a), potentially degradable fraction (b), EDDM, and EDOM (*p* < 0.05). Untreated CSC had the highest soluble fraction (*p* < 0.01) (a), EDDM, and EDOM, which had averages of 49.79 ± 0.13, 94.44 ± 3.34, and 95.69 ± 2.46%, respectively. In contrast, the untreated CSC (raw) had a lower soluble fraction than the potential degradable fraction (b), with an average of 57.17 ± 1.08%. However, there was no interaction effect (*p* > 0.05) on the rate constant (c). There was an effect (*p* < 0.05) of the source of starch on the rate constant (c), with CSC having a higher rate constant than the WBT groups. The modification of starch with the steam method reduced the rate constant (c) by 36.36% when compared with the untreated group (*p* < 0.05).

### 3.3. Gas of Kinetics

The gas kinetic profiles after 96 h of incubation are shown in Table 3 and Figure 1. Negative values were obtained after fitting the raw data to the model [19] for Y = a + b (1−e^(−ct)^). There was an interaction effect (*p* < 0.05) between the source of starch and the starch modification methods in the immediately soluble fraction (a) and the degradation rate constants for the insoluble fraction (c). The WBT treated with the steam process showed lower (*p* < 0.05) than the other groups on the immediately soluble fraction (a). In addition, the results showed that the degradation rate constants for the insoluble fraction (c) in the untreated CSC were significantly higher (*p* < 0.05) than those of the other groups. However, there were no interaction effects on the insoluble fraction of gas production (b) or the potential extent of gas production (|a| + b). The steam method showed greater values of b and |a| + b with 164.00 and 172.42 mL/0.5 g DM, respectively. Furthermore, the cumulative gas production was higher in the untreated group (*p* < 0.01) than in the other groups.

### 3.4. In Vitro Degradability

Table 4 shows the effect of modified starches (CSC and WBT) on IVDMD and IVOMD. According to the results, IVDMD and IVOMD did not exhibit an interaction effect (*p* > 0.05). There were no differences in IVDMD and IVOMD according to the source of starch. The starch modification methods with LA significantly reduced IVDMD at 12 and 24 h of incubation (*p* < 0.05) compared with the untreated group (510.90 and 653.81 g/kg DM, respectively). IVOMD showed the highest value (*p* < 0.05) in the untreated group (679.62 g/kg DM). However, there were no changes in the mean values of IVDMD and IVOMD at 24 h, or in the mean value of IVOMD.

### 3.5. Ruminal pH, and Ammonia-Nitrogen Concentrations

The effects of CSC- and WBT-modified starch on ruminal pH, and NH_3_-N concentrations are shown in Table 5. The CSC and WBT untreated groups showed lower ruminal pH values (*p* < 0.05) than the other groups. In contrast, no interactions were found between pH at 4 h after incubation and NH_3_-N concentrations. The ruminal pH at 4 h of incubation was significantly different (*p* < 0.05) between the starch modification methods, with the lowest value occurring in the untreated group. However, there were no changes in the NH_3_-N concentration. The mean concentrations of NH_3_-N varied from 16.96 ± 1.65 to 18.01 ± 1.40 mg/dL, respectively.

### 3.6. In Vitro Volatile Fatty Acid (VFA) Profiles

Table 6 shows the effect of CSC- and WBT-modified starch on in vitro volatile fatty acid (VFA) profiles, including total VFAs, acetate (C2), propionate (C3), and butyrate (C4) concentrations, as well as the C2:C3 ratio. There were no changes in any parameters due to the dietary treatments. The total VFA had mean values ranging from 95.95 to 94.39 mmol/L.

## 4. Discussion

### 4.1. Chemical Compositions

Cassava chips (CSC) have been used as an energy source in tropical ruminant rations due to their high starch content and more than 90% immediate rumen degradability. However, WBT may eventually replace CSC as a source of energy due to its high starch content. All parameters had no interaction effects between the starch source and the starch modification procedures. This study shows that the source of starch is different in the CSC group, which had lower CP, EE, and ADF values and higher NDF and NFC values than in the untreated WBT group, which is similar to previous reports [5,27]. Generally, CSC contains a DM content of 15.8–23.4%, with 1.85–2.17% CP, 55.0–74.4% NFE, 1.06–1.24% fat, 3.47–3.65% CF, 1.7–2.08% ash, and 78.81–79.20% carbohydrate on a DM basis [27]. Tanzi et al. [5] reported that WBT contains 1.40–2.76% moisture, 17–19% DM CP, 0.21–4.53%DM fat, 2.76–3.0%DM CF, and 25–30% DM carbohydrate content. According to our trial’s findings, the CSC’s NFC content ranged from 75.6% to 77.6%, which is higher than that of WBT, which ranged from 70.9% to 73.0%. Thus, the chemical composition in CSC had lower CP, EE, and ADF values than that of WBT and higher NDF and NFC values than the untreated WBT group.

In this study, the CP content was significantly different based on the modification methods with NaOH treatment, which had negative side effects and resulted in a lower CP value (Table 1). This could be due to the starch granules swelling after exposure to NaOH, resulting in content of essential amino acids (cystine and lysine), amylose, or amylopectin, which creates a continuous gelled matrix that encircles the fragments of starch granules that rupture and leach out of the structure [28,29].

Furthermore, the results showed that starch modification methods increased the ash content when the starch was treated with NaOH and CaOH_2_. The NaOH and CaOH_2_ treatments increased the amount of ash. This is probably due to treatment with alkaline chemicals that contain inorganic components, causing the untreated materials to have more inorganic components [30]. This may have affected the lower content of NFC in the CaOH_2_ group as well. From the table, it can be seen that the GE increased after treatment with steam due to the sample’s rich starch gelatinization and amylopectin concentration. This result is consistent with reports by Pu et al. [31], who discovered this effect with an increased degree of gelatinization in corn starch. It is possible that the processing procedure improves cereal starch through changes in structure and properties, such as granular swelling, crystallite melting, birefringence loss, amylose leaching, and increased amylopectin concentration [31].

### 4.2. In Situ Degradability

This study showed that the interaction between steam treatment and WBT reduced the soluble fractions (a), (b), EDDM, and EDOM in the rumen. Physical modification of starches involves alteration of the starch granules since water and heat only affect them due to the starch-related effects of the increased temperature, which are relatively well documented [8]. By heating in the presence of water, starch undergoes a structural transformation and gelatinization, its crystal structure disintegrates, and its solubility and rheological properties are modified. With dry superheating, a high temperature also produces modified starches that are soluble and swellable in cold water, thus decreasing the soluble fractions (a), EDDM, and EDOM while increasing the potentially degradable fraction (b) [9,28]. This could be due to the fact that the starch’s crystalline structure might shift.

Narwojsz et al. [32] found that these effects occur because steaming adds water to the feed, which makes it more likely to be broken down by enzymes. Then, when the starch cools, it undergoes retrogradation, which is a process that crystallizes the starch and enhances its resistance to digesting enzymes [32,33]. Typically, most of the feed’s DM is composed of starch, which affects the degradation rate constant (c) and effective degradability (DM and OM), which are positively correlated in a high-starch feed [12].

In this study, the source of starch was significantly different. The WBT showed a lower degradation rate constant (c) and starch content (NFC) than the CSC. A cereal grain’s pericarp may cause or reflect variations in the starch’s molecular makeup—specifically, the ratio of amylose to amylopectin, with the availability of additional feed components in the grain, such as fat, fiber, and protein, which can form bonds with starch molecules [34,35]. This may have an effect on the lower degradation rate constant (c) value.

In addition, when considering only the influence of starch modification methods, the steam method reduced the rate constant (c) due to physical treatment affecting the various properties of starch granules, which may have a significant impact on isolated amylolytic enzymes’ capacity to digest starch and, consequently, on the rumen’s ability to degrade starch [36]. Furthermore, the high temperatures attained in our experiment may have resulted in reduced in situ solubility and degradability. This could be due to the gelatinization and debranching of amylopectin, which led to more amylose in the starch. When the starch was re-cooled and gelatinized at high temperatures, the amylose broke down, which made the starch less soluble and easier to break down in the rumen [37].

This study demonstrated a markedly lower soluble fraction (a), (b), EDDM, and EDOM in the steam-treated group than in the untreated group. The results of this experiment showed that steam treatment decreased the solubility (a) of CSC and WBT and their ED in the rumen. As a result of our findings, EDDM and EDDOM significantly reduced DM and OM loss in the rumen when physical treatment was applied to CSC [28]. According to Karami et al. [38], the protein matrix surrounding the starch granules in the endosperm is disrupted, and the starch granules themselves are disorganized, which leads to increased starch degradability in steamed grains. Furthermore, high steaming temperatures and alkalosis may harm the protective physical shell of corn grains, reducing their ability to dissolve in water and the rate at which they are digested by microorganisms in the rumen [36].

### 4.3. Kinetics of Gas

The winged bean tuber (WBT) has a high starch content, making it comparable to cassava chips in terms of producing specific useful gases and digestion. In this study, the immediately soluble fraction (a) had a negative value, which is a deviation from the exponential result of fermentation or a delay in the onset of fermentation. This is due to the fact that when the soluble portion of the substrate has been consumed, but fermentation of the cell walls has not yet begun, there is a lag time before microbial colonization occurs [11]. Thus, the absolute value |a| was used in the mathematical model that ideally describes gas production kinetics and indicates the soluble fraction’s fermentation [32].

In the present study, the immediately soluble fraction (a) and the degradation rate constants for the insoluble fraction (c) decreased in WBT treated by steaming, but there was greater gas production from the insoluble fraction (b) and the potential extent of gas production (|a| + b). This was most likely due to the resistant starch of steamed WBT, which also resulted in a lower fermentable soluble fraction that fermented slower [11]. Starch changes structurally and gelatinizes when heated in the presence of water. Its crystal structure also disintegrates, and its solubility and rheological properties are altered [39]. When combined with modification methods for starches, the difference in particle size, fiber content, pelletizing, extruding, pressure toasting, and other feed production techniques may significantly alter how starch in compound feeds breaks down in the rumen [40].

The cumulative gas at 96 h of incubation was lower in the steamed and LA-treated groups. The benefits of physically processing grain have been identified with the use of steam treatments, which are typically accomplished through derivatization processes such as etherification, esterification, and cross-linking [41]. In physically treating grain, hydrophilic hydroxyl groups are formed, and the formation of hydrogen bonds between hydroxyl groups and water molecules is inhibited by the increased hydrophobicity of starch [42].

Since aqueous starch dispersion has the propensity to thicken upon chilling, Huang et al. [34] showed that the interaction of amylose molecules leads to gelatinization because of the outer layers and a grain’s physical composition, which are frequently impermeable to water and bacterial breakdown. These factors influence the digestibility of starch in the gastrointestinal system. Furthermore, in vitro studies have shown that LA treatment slows the enzymatic activity of grain amylases, resulting in less starch degradation [43]. Exactly how LA affects the structure of starch is not yet entirely known. According to one theory, LA limits enzymatic attack by linearizing the branching amylopectin-molecule [44].

Another theory is that interactions between gluten and LA can function as barriers to enzymatic breakdown. LA treatment has the ability to slow the enzymatic action of amylases in grains [42]. Moreover, Deckardt et al. [39] demonstrated that, compared to dry-rolled barley, the gas volume was reduced early on, and the gas production constant rate increased after 24 h of steeping barley in water containing 7.5 g of LA.

### 4.4. In Vitro Degradability

The WBT and CSC both have a high starch content, which leads to a high degradation rate. Controlling the rate of degradation may provide more opportunities for rumen microbes to use starch for growth. In this study, IVDMD at 12 and 24 h decreased with modified starch produced by NaOH, CaOH_2,_ and LA treatment when compared to no treatment. The modification methods significantly changed the IVDMD and IVOMD, with NaOH, CaOH_2_, and LA treatments influencing in vitro degradability compared to the untreated group.

Rojas et al. [45] demonstrated that sorghum treated with NaOH had lower total apparent degradability when measured in the entire gastrointestinal tract, which impacts low feed utilization in animals. This could be due to some physicochemical changes, such as starch granule enlargement that occurs after NaOH treatment and amylose and amylopectin changing the structure and outer layers. These processes affect the digestibility of starch in the gastrointestinal tract, such that it is often impermeable to water and bacterial degradation [46]. In addition, this mechanism may be explained by the effect of CaOH_2_ on reduced in vitro degradability [47].

Treatment with LA impacts starch degradation rates. This probably has to do with the fact that LA causes linearization of the branching amylopectin molecule [39], which makes amylopectin more like amylose and operates as a barrier to enzymatic digestion. Additionally, restricting enzymatic attack is a potential interaction between LA and gluten [9]. Shaikh et al. [48] demonstrated that treatment with LA increased the resistant starch content and esterified the hydroxyl groups in all places, generating bulky chains and causing steric hindrance to digestive enzymes. Similarly, Pilachai et al. [49] found that the DM digestion rates of starch in cassava meal treated with 1% LA were slowed under in vitro conditions. Furthermore, Khol-Parisini et al. [14] demonstrated that treating barley grain with LA slowed the rate at which barley starch was fermented.

In this study, treatment with LA and CaOH_2_ resulted in a lower IVOMD at 12 h. It is possible that the resistant starch treatment effect of LA lowers IVDMD, which may result in lower IVOMD [50]. In addition, this is probably due to the treatment with alkaline chemicals (CaOH_2_) that contain inorganic components, causing the replacement of organic components and possibly making the content of IVOMD lower in the CaOH_2_ group as well [51].

### 4.5. Ruminal pH and Ammonia-Nitrogen Concentrations

The rapid degradation of starch in WBT and CSC resulted in a decrease in ruminal pH. Rapidly lowering rumen pH may harm rumen microbes and reduce their ability to digest feed. Ruminal pH ranged from 6.93–7.16, which is in the normal range of rumen ecology (pH 6.20–7.00) [13]. In the present study, different modification methods influenced the ruminal pH at 2 and 4 h of incubation time, and the lowest value occurred in the untreated group. The rapid and higher in vitro degradation of the untreated samples compared with the starch modification methods may explain the reduction in ruminal pH observed in the untreated group treatment. The rapid degradation of starch in the untreated group (without modified starch) is due to higher microbial fermentation, resulting in increased acid production and a lower pH value [8].

Humer and Zebeli [44] reported that a slow-fermented diet in the rumen results in a higher ruminal pH, which might be due to the pH-raising effect of the alkali treatments, as well as improved rumen fermentation. Srakaew et al. [8] reported that slow-fermented starch with alkaline treatment produced fewer short-chain fatty acids (SCFA) and had a higher pH. Likewise, Chanjula et al. [10] reported that the pH of the grain treated with CaOH_2_ was higher than that of a control group. Deckardt et al. [42] reported that the ruminal pH value of barley treated with LA was higher than that of the control group. Similarly, Pilachai et al. [4] showed that the LA group had a higher ruminal pH than the SARA threshold (ruminal pH 5.8).

In this study, the NH_3_-N values were within the normal range [10,52]. It has been reported that the optimal level is 15–30 mg/dL to support microbial protein synthesis and optimize ruminal feed digestibility. However, there was no effect on the NH_3_-N concentration. This could be due to the fact that the modified starch did not affect the rate of degradability of protein [8].

### 4.6. In Vitro VFA Profiles

WBT may serve as a substrate for the production of VFA, which is comparable to cassava root because of its high starch content [1,53]. In this study, there was no change in the total VFA or VFA profiles. This could be due to CSC and WBT containing starches that have solubility and are rapidly degradable in the rumen, thus causing a similar fermentation process within the rumen [4,54]. Beckett et al. [55] found that disparities might be explained by many factors that affect rumen VFA, such as forage source, rumen volume, flow rate, animal-to-animal variation, interactions between these factors, and dietary particle size. In addition, it is probably affected by the length of incubation, species of the animal donor, chemical composition, and biological activity of microbial factors, which did not alter overall VFA synthesis [56]. Similarly, Srakaew et al. [8] discovered that increasing the amount of modified starch in CSC and corn seed had no impact on the production of C2, C3, and C4.

## 5. Conclusions

This study demonstrated that WBT has a greater CP and EE content but a lower NDF content when compared with CSC. Treatment with NaOH decreased the CP content. The WBT treated with steam had a lower soluble fraction (a) and EDDM in situ. Furthermore, steaming methods showed a lower degradation rate than in situ methods.

Compared to other methods, no treatment resulted in higher cumulative gas production, IVDMD, or IVOMD at 12 h but lowered the pH in vitro. The source of starch (CSC or WBT) and starch modification methods did not influence the in vitro fermentation end-product (VFA). In conclusion, considering that WBT contains more starch and protein than CSC, steaming WBT might be a more successful strategy to enhance feed efficiency, decrease degradability, and maintain rumen pH, resulting in a convenient alternative to CSC. However, further in vivo research needs to be done to determine the actual intestinal availability of steam-treated WBT in animal production.

## Figures and Tables

**Figure 1 animals-13-01640-f001:**
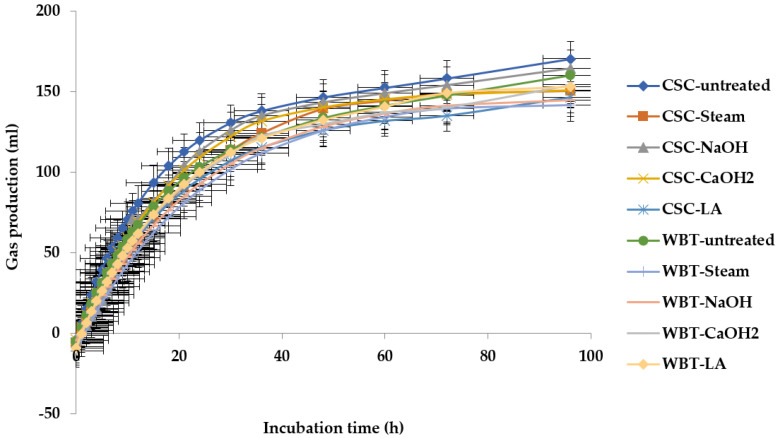
Cumulative gas production curves of cassava chip and winged bean tuber treated with various modified starch methods. CSC-untreated = Cassava chip-untreated, CSC-Steam = Cassava chip-steam treated, CSC-NaOH = Cassava chip-sodium hydroxide treated, CSC-CaOH_2_ = Cassava chip-calcium hydroxide treated, CSC-LA = Cassava chip-lactic acid treated, WBT-untreated = Winged bean tuber-untreated, WBT-Steam = Winged bean tuber-steam treated, WBT-NaOH = Winged bean tuber-sodium hydroxide treated, WBT-CaOH_2_ = Winged bean tuber-calcium hydroxide treated, WBT-LA = Winged bean tuber- lactic acid treated.

**Table 1 animals-13-01640-t001:** The comparison of chemical compositions in cassava chip- and winged bean tuber-modified starch.

Items		DM (g/kg)	CP	EE	NDF	ADF	Ash	NFC	GE (kcal/g DM)
			(g/kg of DM)						
CSC	Raw	909.72 ± 9.44	20.42 ± 0.40	7.66 ± 0.82	275.00 ± 7.28	40.11 ± 0.09	29.45 ± 0.49	667.46 ± 7.34	3.73 ± 0.05
	Steam	913.20 ± 10.24	19.47 ± 0.08	7.71 ± 0.35	271.09 ± 7.82	37.05 ± 0.14	28.86 ± 0.53	672.88 ± 6.85	3.76 ± 0.06
	NaOH	912.27 ± 0.62	18.14 ± 0.15	7.53 ± 0.40	267.59 ± 3.65	37.77 ± 0.63	57.86 ± 5.57	648.88 ± 1.66	3.75 ± 0.05
	CaOH_2_	905.99 ± 3.12	19.44 ± 0.47	7.55 ± 0.01	269.05 ± 5.59	31.41 ± 2.19	65.23 ± 0.12	638.74 ± 4.99	3.72 ± 0.07
	LA	900.98 ± 5.31	20.29 ± 0.69	7.67 ± 0.58	271.40 ± 5.06	32.23 ± 0.38	27.74 ± 1.99	672.90 ± 4.33	3.74 ± 0.02
WBT	Raw	906.70 ± 1.49	190.25 ± 0.75	12.66 ± 2.03	233.70 ± 4.05	59.43 ± 2.00	32.33 ± 1.44	531.06 ± 4.21	3.78 ± 0.03
	Steam	917.94 ± 0.76	183.66 ± 0.34	11.77 ± 1.51	239.69 ± 3.74	63.02 ± 2.80	31.75 ± 0.31	533.12 ± 5.91	3.79 ± 0.03
	NaOH	902.89 ± 2.10	177.97 ± 4.28	11.70 ± 1.07	227.64 ± 5.95	62.71 ± 0.86	55.58 ± 8.78	527.11 ± 2.53	3.78 ± 0.05
	CaOH_2_	906.61 ± 0.95	189.06 ± 2.00	11.84 ± 0.35	228.62 ± 5.79	61.02 ± 4.08	61.71 ± 1.66	508.76 ± 5.78	3.77 ± 0.07
	LA	908.39 ± 8.17	184.94 ± 2.31	11.86 ± 1.47	224.36 ± 5.83	63.29 ± 4.11	30.94 ± 2.30	547.90 ± 7.28	3.78 ± 0.06
SEM		6.54	1.70	0.11	8.66	2.56	3.50	3.46	0.008
*p*-value		0.60	0.05	0.97	0.84	0.16	0.78	0.06	0.41
Source of starch	CSC	908.43 ± 4.49	19.55 ^b^ ± 0.81	7.62 ^b^ ± 0.07	270.83 ^a^ ± 2.51	35.71 ^b^ ± 3.35	41.83 ± 16.27	660.17 ^a^ ± 13.88	3.74 ± 0.06
	WBT	908.51 ± 5.05	185.18 ^a^ ± 436	11.97 ^a^ ± 0.35	230.80 ^b^ ± 5.36	61.89 ^a^ ± 1.46	42.46 ± 13.36	529.59 ^b^ ± 12.57	3.78 ± 0.05
Starch modification methods	Raw	908.21 ± 1.51	105.34 ^a^ ± 84.91	10.16 ± 2.50	254.35 ± 2.65	49.77 ± 9.66	30.89 ^b^ ± 1.44	599.26 ^ab^ ± 68.20	3.74 ^b^ ± 0.03
	Steam	915.57 ± 2.37	101.56 ^ab^ ± 82.10	9.74 ± 2.03	255.39 ± 1.57	50.03 ± 12.98	30.31 ^b^ ± 1.45	603.00 ^ab^ ± 69.88	3.77 ^a^ ± 0.08
	NaOH	907.58 ± 4.69	98.06 ^b^ ± 79.91	9.61 ± 2.08	247.62 ± 1.99	50.24 ± 12.47	56.72 ^a^ ± 1.14	588.00 ^bc^ ± 60.89	3.75 ^b^ ± 0.07
	CaOH_2_	906.30 ± 0.31	104.25 ^a^ ± 84.81	9.70 ± 2.15	248.83 ± 2.35	46.22 ± 14.80	63.47 ^a^ ± 1.76	573.75 ^c^ ± 64.99	3.76 ^ab^ ± 0.05
	LA	904.68 ± 3.70	102.62 ^a^ ± 82.32	9.76 ± 2.10	247.88 ± 2.35	47.76 ± 15.53	29.34 ^b^ ± 1.60	610.40 ^a^ ± 62.50	3.76 ^ab^ ± 0.03

^a–c^ Mean in the same row with different superscript differ (*p* < 0.05), SEM = standard error of the mean, interaction = source of starch × modified methods, DM = dry matter, CP = crude protein, EE = ether extract, CF = crude fiber, NDF = neutral detergent fiber, ADF = acid detergent fiber, NFC = non-fiber carbohydrates, GE = gross energy, CSC = cassava chip, WBT = winged bean tuber, Raw = raw stuff (untreated), LA = lactic acid, NaOH = Sodium hydroxide treated, CaOH_2_ = Calcium hydroxide treated.

**Table 2 animals-13-01640-t002:** In situ degradation characteristics of cassava chip and winged bean tuber-modified starch.

Items		In Situ Degradation Characteristics			Effective Dry Matter Degradability (%)	Effective Organic Matter Degradability (%)
		a (%)	b (%)	c (h^−1^)		
CSC	Raw	49.79 ^a^ ± 0.13	57.17 ^d^ ± 1.08	0.20 ± 0.07	94.44 ^a^ ± 3.34	95.69 ^a^ ± 2.46
	Steam	11.12 ^f^ ± 1.09	67.90 ^a^ ± 1.83	0.09 ± 0.01	54.37 ^d^ ± 2.99	74.49 ^c^ ± 1.78
	NaOH	13.95 ^de^ ± 0.56	64.92 ^bc^ ± 0.56	0.17 ± 0.01	64.03 ^c^ ± 0.056	75.40 ^c^ ± 0.91
	CaOH_2_	14.99 ^cd^ ± 0.32	65.71 ^bc^ ± 0.57	0.18 ± 0.04	65.65 ^c^ ± 2.93	75.81 ^c^ ± 1.03
	LA	16.17 ^c^ ± 0.82	64.07 ^c^ ± 1.22	0.18 ± 0.02	65.94 ^c^ ± 1.32	75.74 ^c^ ± 0.62
WBT	Raw	21.93 ^b^ ± 0.93	65.51 ^bc^ ± 1.84	0.19 ± 0.01	73.95 ^b^ ± 0.37	84.22 ^b^ ± 0.94
	Steam	5.73 ^g^ ± 1.02	63.94 ^c^ ± 3.91	0.06 ± 0.04	37.92 ^e^ ± 6.24	73.71 ^c^ ± 2.31
	NaOH	12.27 ^f^ ± 0.16	65.54 ^bc^ ± 1.17	0.11 ± 0.00	57.56 ^d^ ± 0.55	73.99 ^c^ ± 2.50
	CaOH_2_	11.89 ^f^ ± 0.78	66.05 ^bc^ ± 1.24	0.10 ± 0.02	55.98 ^d^ ± 1.71	75.03 ^c^ ± 1.65
	LA	12.53 ^ef^ ± 0.36	65.37 ^bc^ ± 1.39	0.13 ± 0.02	57.63 ^d^ ± 1.78	76.15 ^c^ ± 2.41
SEM		0.89	1.67	0.06	2.57	1.77
*p*-value		<0.01	<0.01	0.57	<0.01	<0.01
Source of starch	CSC	21.20 ± 14.39	63.95 ± 3.62	0.16 ^a^ ± 0.04	68.89 ± 13.47	79.01 ± 8.37
	WBT	12.73 ± 5.19	64.97 ± 0.71	0.14 ^b^ ± 0.04	55.59 ± 11.42	76.00 ± 3.66
	*p*-value	<0.01	<0.01	<0.01	<0.01	<0.01
Starch modification methods	Raw	35.86 ± 13.39	61.34 ± 4.17	0.22 ^a^ ± 0.00	84.20 ± 10.24	89.96 ± 5.74
	Steam	8.43 ± 2.69	65.92 ± 1.98	0.09 ^c^ ± 0.01	46.14 ± 8.23	75.15 ± 0.66
	NaOH	13.11 ± 0.84	65.23 ± 0.31	0.13 ^b^ ± 0.03	60.79 ± 3.24	74.69 ± 0.71
	CaOH_2_	13.44 ± 1.55	65.88 ± 0.17	0.14 ^b^ ± 0.04	60.82 ± 4.84	74.37 ± 0.66
	LA	14.35 ± 1.82	64.72 ± 0.65	0.14 ^b^ ± 0.02	61.78 ± 4.16	75.95 ± 0.20
	*p*-value	<0.01	<0.01	<0.01	<0.01	<0.01

^a–g^ Mean in the same row with different superscript differ (*p* < 0.05), SEM = standard error of the mean, interaction = source of starch × modified methods, CSC = cassava chip, WBT = winged bean tuber, Raw = raw stuff (untreated), NaOH = Sodium hydroxide treated, CaOH_2_ = Calcium hydroxide treated, LA = LA = lactic acid treated, a = a soluble fraction; b = potentially degradable fraction (%), c = degradation rate constant b (h^−1^).

**Table 3 animals-13-01640-t003:** The effect of cassava chip and winged bean tuber-modified starch on gas kinetics.

Items		Kinetics of Gas			
		a	b	c	|a| + b
CSC	Raw	−3.50 ^a^ ± 0.71	157.40 ± 6.64	0.064 ^a^ ± 0.00	160.90 ± 5.78
	Steam	−6.00 ^ab^ ± 1.63	169.58 ± 7.05	0.041 ^d^ ± 0.00	175.58 ± 7.46
	NaOH	−3.66 ^a^ ± 0.18	141.36 ± 1.91	0.051 ^c^ ± 0.00	163.03 ± 1.74
	CaOH_2_	−3.78 ^a^ ± 0.56	163.77 ± 3.17	0.049 ^c^ ± 0.00	167.55 ± 2.06
	LA	−5.53 ^a^ ± 1.02	144.50 ± 4.35	0.050 ^c^ ± 0.00	150.03 ± 5.35
WBT	Raw	−4.57 ^a^ ± 0.95	144.69 ± 9.55	0.057 ^b^ ± 0.00	149.26 ± 1.33
	Steam	−10.84 ^d^ ± 1.15	158.41 ± 1.48	0.041 ^d^ ± 0.00	169.25 ± 1.89
	NaOH	−5.32 ^a^ ± 1.50	153.83 ± 3.88	0.042 ^d^ ± 0.00	159.15 ± 4.79
	CaOH_2_	−9.11 ^cd^ ± 1.03	153.49 ± 6.60	0.051 ^d^ ± 0.00	162.59 ± 5.73
	LA	−8.11 ^bc^ ± 0.90	154.30 ± 2.47	0.050 ^d^ ± 0.00	162.42 ± 3.46
SEM		0.93	5.84	0.002	6.90
*p*-value		<0.01	0.10	<0.01	0.14
Source of starch	CSC	−4.49 ± 1.05	155.32 ± 1.68	0.051±	163.42 ± 1.54
	WBT	−8.34 ± 2.34	155.01 ± 1.91	0.048±	163.35 ± 1.99
	*p*-value	0.05	0.16	<0.01	0.06
Starch modification methods	Raw	−4.49 ± 0.53	147.59 ^b^ ± 1.55	0.060 ± 0.01	155.08 ^b^ ± 7.03
	Steam	−8.42 ± 2.42	164.00 ^a^ ± 6.21	0.041 ± 0.01	172.42 ^a^ ± 2.55
	NaOH	−4.49 ± 0.83	147.59 ^b^ ± 9.00	0.046 ± 0.00	161.09 ^ab^ ± 9.96
	CaOH_2_	−6.45 ± 2.66	158.63 ^ab^ ± 9.34	0.050 ± 0.00	165.07 ^ab^ ± 3.35
	LA	−6.82 ± 1.29	149.40 ^b^ ± 1.16	0.050 ± 0.00	156.22 ^b^ ± 0.62
	*p*-value	<0.01	<0.01	<0.01	<0.01

^a–d^ Mean in the same row with different superscript differ (*p* < 0.05), SEM = standard error of the mean, interaction = source of starch × modified methods, CSC = cassava chip, WBT = winged bean tuber, NaOH = Sodium hydroxide treated, CaOH_2_ = Calcium hydroxide treated, a = the gas production from the immediately soluble fraction (mL); b = the gas production from the insoluble fraction (mL); c = the gas production rate constant for the insoluble fraction (mL h^−1^); |a| + b, the gas potential extent of gas production.

**Table 4 animals-13-01640-t004:** The effect of cassava chip and winged bean tuber-modified starch on in vitro dry matter degradability and in vitro organic matter degradability.

Items		IVDMD (g/kg DM)			IVOMD (g/kg DM)		
		12 h	24 h	Means	12 h	24 h	Means
CSC	Raw	548.60 ± 3.40	677.60 ± 0.80	613.10 ± 4.50	748.70 ± 0.90	838.40 ± 2.80	793.55 ± 9.95
	Steam	532.20 ± 3.20	666.50 ± 1.10	599.35 ± 7.15	735.80 ± 0.80	825.80 ± 3.45	780.80 ± 5.00
	NaOH	519.60 ± 6.40	661.90 ± 0.90	590.75 ± 1.15	725.90 ± 4.90	831.80 ± 5.00	778.85 ± 2.95
	CaOH_2_	511.70 ± 4.10	661.10 ± 0.30	586.40 ± 4.70	719.50 ± 8.50	834.50 ± 2.10	777.00 ± 7.50
	LA	504.40 ± 2.20	651.97 ± 6.93	578.19 ± 3.79	699.80 ± 8.80	808.60 ± 10.60	754.20 ± 9.30
WBT	Raw	546.30 ± 3.50	681.64 ± 7.04	613.97 ± 7.67	738.40 ± 7.36	854.90 ± 9.60	796.65 ± 0.90
	Steam	536.20 ± 3.20	670.80 ± 1.80	603.50 ± 7.30	712.90 ± 2.90	838.90 ± 9.30	775.90 ± 3.00
	NaOH	519.80 ± 3.00	666.70 ± 2.90	593.25 ± 3.45	720.60 ± 6.40	847.50 ± 10.30	784.05 ± 3.45
	CaOH_2_	521.50 ± 4.10	664.62 ± 5.56	593.06 ± 1.56	691.50 ± 2.50	845.90 ± 9.30	768.70 ± 7.20
	LA	517.40 ± 8.80	655.64 ± 10.16	586.52 ± 9.12	688.80 ± 6.40	820.20 ± 1.10	754.50 ± 3.05
SEM		12.15	6.93	12.63	11.21	11.72	10.09
*p*-value		0.81	0.74	0.71	0.97	0.96	0.59
Source of starch	CSC	523.30 ± 5.65	663.81 ± 8.35	593.56 ± 11.92	725.94 ± 6.37	827.82 ± 10.45	776.88 ± 4.07
	WBT	528.24 ± 1.17	667.88 ± 8.48	598.06 ± 9.63	710.44 ± 8.54	841.48 ± 11.79	775.96 ± 5.41
	*p*-value	0.86	0.53	0.78	0.18	0.38	0.95
Starch modification methods	Raw	547.45 ^a^ ± 1.15	679.62 ^a^ ± 0.02	613.54 ± 0.44	743.55 ^a^ ± 5.15	846.65 ± 5.80	795.10 ± 0.32
	Steam	534.20 ^ab^ ± 2.00	668.65 ^ab^ ± 2.15	601.43 ± 2.07	724.35 ^b^ ± 11.45	832.35 ± 6.55	778.35 ± 2.45
	NaOH	519.70 ^bc^ ± 0.10	664.30 ^b^ ± 2.40	592.00 ± 1.25	723.25 ^b^ ± 2.65	839.65 ± 7.85	781.45 ± 2.60
	CaOH_2_	516.60 ^bc^ ± 4.90	662.86 ^b^ ± 1.76	589.73 ± 3.33	705.50 ^c^ ± 14.00	840.20 ± 5.70	772.85 ± 4.15
	LA	510.90 ^c^ ± 6.50	653.81 ^c^ ± 1.84	582.35 ± 4.17	694.30 ^c^ ± 5.50	814.40 ± 8.25	754.35 ± 1.37
	*p*-value	<0.01	<0.01.	0.98	<0.01	0.12	1.00

^a–c^ Mean in the same row with different superscript differ (*p* < 0.05), SEM = standard error of the mean, interaction = source of starch × modified methods, CSC = cassava chip, WBT = winged bean tuber, NaOH = Sodium hydroxide treated, CaOH_2_ = Calcium hydroxide treated, IVDMD = in vitro dry matter degradability, IVOMD = in vitro organic matter degradability.

**Table 5 animals-13-01640-t005:** The effect of cassava chip- and winged bean tuber-modified starch on ruminal pH, and ammonia-nitrogen (NH_3_-N).

Items		pH	NH_3_-N (mg/dL)		
		4 h	2 h	4 h	Means
CSC	Raw	6.86 ± 0.02	16.58 ± 1.17	19.42 ± 0.01	18.00 ± 1.42
	Steam	6.90 ± 0.03	15.31 ± 0.10	18.61 ± 0.40	16.96 ± 1.65
	NaOH	6.94 ± 0.01	15.61 ± 0.20	18.91 ± 0.70	17.26 ± 1.65
	CaOH_2_	7.06 ± 0.00	16.01 ± 0.60	18.41 ± 0.40	17.21 ± 1.20
	LA	6.93 ± 0.00	15.81 ± 0.20	18.92 ± 0.30	17.36 ± 1.56
WBT	Raw	6.96 ± 0.00	16.61 ± 0.30	19.91 ± 0.20	18.01 ± 1.40
	Steam	6.96 ± 0.00	16.01 ± 0.40	19.22 ± 0.40	17.61 ± 1.60
	NaOH	6.99 ± 0.00	16.31 ± 0.10	18.01 ± 0.30	17.41 ± 1.10
	CaOH_2_	7.02 ± 0.01	16.55 ± 0.80	18.79 ± 0.54	17.67 ± 1.12
	LA	6.97 ± 0.01	16.05 ± 0.60	18.61 ± 0.16	17.33 ± 1.28
SEM		0.03	0.55	0.39	1.41
*p* = value		0.32	0.96	0.65	1.00
Source of starch	CSC	6.93 ± 0.07	15.86 ± 0.43	18.85 ± 0.34	17.36 ± 0.34
	WBT	6.98 ± 0.06	16.31 ± 0.25	18.91 ± 0.35	17.61 ± 0.24
	*p*-value	0.21	0.23	0.83	0.79
Starch modification methods	Raw	6.85 ^c^ ± 0.01	16.59 ± 0.20	19.41 ± 0.00	18.00 ± 0.01
	Steam	6.94 ^b^ ± 0.03	15.66 ± 0.35	18.92 ± 0.30	17.29 ± 0.33
	NaOH	6.96 ^b^ ± 0.02	15.96 ± 0.35	18.71 ± 0.20	17.34 ± 0.08
	CaOH_2_	7.04 ^a^ ± 0.02	16.28 ± 0.27	18.60 ± 0.19	17.44 ± 0.23
	LA	6.95 ^b^ ± 0.02	15.93 ± 0.12	18.76 ± 0.16	17.35 ± 0.02
	*p*-value	<0.01	0.52	0.32	0.98

^a–c^ Mean in the same row with different superscript differ (*p* < 0.05), SEM = standard error of the mean, interaction = source of starch × modified methods, CSC = cassava chip, WBT = winged bean tuber, NaOH = Sodium hydroxide treated, CaOH_2_ = Calcium hydroxide treated.

**Table 6 animals-13-01640-t006:** The effect of cassava chip- and winged bean tube-modified starch on in vitro volatile fatty acid (VFA) profiles.

Items		Total VFA (mmol/L)			C2, %			C3, %			C4, %			C2 to C3 Ratio		
		2 h	4 h	Mean	2 h	4 h	Mean	2 h	4 h	Mean	2 h	4 h	Mean	2 h	4 h	Mean
CSC	Raw	91.97 ± 1.02	96.81 ± 2.15	94.39 ± 0.12	66.90 ± 0.15	66.56 ± 0.06	66.73 ± 0.10	21.99 ± 0.11	22.84 ± 0.04	22.42 ± 0.02	11.110 ± 0.15	10.60 ± 0.13	10.85 ± 0.24	3.04 ± 0.02	2.91 ± 0.02	2.98 ± 0.18
	Steam	90.84 ± 0.40	95.09 ± 1.30	92.96 ± 0.38	66.81 ± 0.23	66.64 ± 0.05	66.73 ± 0.09	21.82 ± 0.14	22.31 ± 0.05	22.06 ± 0.04	11.37 ± 0.18	11.05 ± 0.14	11.21 ± 1.25	3.06 ± 0.03	2.99 ± 0.04	3.02 ± 0.15
	NaOH	90.37 ± 0.80	96.08 ± 2.05	93.22 ± 0.52	66.70 ± 0.18	66.74 ± 0.18	66.72 ± 0.07	21.34 ± 0.21	22.11 ± 0.02	21.73 ± 0.02	11.96 ± 0.10	11.15 ± 0.05	11.55 ± 1.62	3.13 ± 0.04	3.02 ± 0.03	3.07 ± 0.12
	CaOH_2_	90.40 ± 0.65	95.88 ± 0.35	93.14 ± 0.63	67.09 ± 0.20	66.79 ± 0.14	66.94 ± 0.26	22.09 ± 0.18	22.16 ± 0.05	22.12 ± 0.05	10.82 ± 0.08	11.05 ± 0.08	10.94 ± 0.17	3.04 ± 0.05	3.01 ± 0.04	3.03 ± 0.07
	LA	90.55 ± 0.45	95.19 ± 0.40	92.87 ± 0.45	67.58 ± 0.25	65.93 ± 0.26	66.76 ± 0.13	21.53 ± 0.10	22.67 ± 0.03	22.10 ± 0.06	10.89 ± 0.07	11.40 ± 0.10	11.14 ± 0.14	3.14 ± 0.05	2.91 ± 0.04	3.02 ± 0.08
WBT	Raw	91.88 ± 0.30	96.73 ± 0.60	94.30 ± 0.85	66.87 ± 015	66.91 ± 0.52	66.89 ± 0.70	21.90 ± 0.21	22.01 ± 0.04	21.96 ± 0.02	11.23 ± 0.10	11.08 ± 0.16	11.15 ± 0.21	3.05 ± 0.07	3.04 ± 0.03	3.05 ± 0.11
	Steam	90.61 ± 0.10	94.84 ± 0.25	92.72 ± 0.15	66.53 ± 0.16	66.24 ± 0.32	66.39 ± 0.40	22.26 ± 0.09	22.74 ± 0.05	22.50 ± 0.05	11.21 ± 0.12	11.02 ± 0.02	11.11 ± 0.90	2.99 ± 0.06	2.91 ± 0.02	2.95 ± 0.10
	NaOH	90.36 ± 0.25	96.00 ± 0.50	93.18 ± 0.35	66.56 ± 0.20	66.13 ± 0.11	66.35 ± 0.53	21.78 ± 0.10	22.78 ± 0.05	22.28 ± 0.04	11.66 ± 0.15	11.09 ± 0.03	11.37 ± 1.02	3.06 ± 0.04	2.90 ± 0.05	2.98 ± 0.09
	CaOH_2_	91.08 ± 0.20	94.82 ± 0.40	92.95 ± 0.48	66.35 ± 0.14	66.22 ± 0.18	66.29 ± 0.47	22.01 ± 0.12	22.17 ± 0.02	22.09 ± 0.03	11.64 ± 0.20	11.61 ± 0.17	11.62 ± 1.52	3.01 ± 0.05	2.99 ± 0.05	3.00 ± 0.14
	LA	91.04 ± 0.30	95.05 ± 0.75	93.04 ± 0.95	66.96 ± 0.28	66.14 ± 0.10	66.55 ± 0.18	21.85 ± 0.15	22.79 ± 0.03	22.32 ± 0.03	11.19 ± 0.06	11.07 ± 0.04	11.13 ± 1.74	3.06 ± 0.04	2.90 ± 0.03	2.98 ± 0.08
SEM		1.35	0.71	2.39	0.41	0.40	0.41	0.36	0.25	0.38	0.41	0.34	0.39	0.44	0.29	0.07
*p*-value		0.99	0.94	0.89	0.69	0.69	0.69	0.54	0.14	0.87	0.39	0.64	0.88	0.81	0.81	0.36
Source of starch	CSC	90.83 ± 0.50	95.81 ± 0.15	93.32 ± 1.85	67.02 ± 0.30	66.53 ± 0.10	66.78 ± 0.30	21.75 ± 0.10	22.42 ± 0.03	22.09 ± 0.04	11.23 ± 0.07	11.05 ± 0.16	11.14 ± 1.55	3.08 ± 0.06	2.97 ± 0.04	3.02 ± 0.05
	WBT	90.99 ± 0.54	95.49 ± 0.25	93.24 ± 2.62	66.65 ± 0.12	66.33 ± 0.16	66.49 ± 0.24	21.96 ± 0.12	22.50 ± 0.05	22.23 ± 0.02	11.39 ± 0.08	11.17 ± 0.10	11.28 ± 1.25	3.04 ± 0.02	2.95 ± 0.05	2.99 ± 0.04
*p* value		0.85	0.49	0.96	0.50	0.38	0.16	0.41	0.80	0.53	0.83	0.67	0.74	0.30	0.56	0.41
Starch modification methods	Raw	91.93 ± 0.06	96.77 ± 1.30	94.35 ± 0.20	66.89 ± 0.18	66.74 ± 0.27	66.81±	21.95 ± 0.14	22.43 ± 0.06	22.19 ± 0.02	11.17 ± 0.20	10.84 ± 0.04	11.00 ± 1.95	3.05 ± 0.02	2.98±	3.01 ± 0.03
	Steam	90.73 ± 0.15	94.97 ± 1.74	92.84 ± 0.40	66.67 ± 0.20	66.44 ± 0.25	66.56±	22.04 ± 0.20	22.53 ± 0.05	22.28 ± 0.02	11.29 ± 0.16	11.04 ± 0.08	11.16 ± 1.65	3.03 ± 0.03	2.95 ± 0.02	2.99 ± 0.03
	NaOH	90.37 ± 0.20	96.04 ± 3.58	93.20 ± 0.30	66.63 ± 0.23	66.44 ± 0.41	66.54±	21.56 ± 0.11	22.45 ± 0.06	22.01 ± 0.03	11.81 ± 0.14	11.12 ± 0.10	11.46 ± 1.47	3.09 ± 0.05	2.96 ± 0.04	3.03 ± 0.02
	CaOH_2_	90.74 ± 0.15	95.35 ± 1.05	93.05 ± 0.20	66.72 ± 0.14	66.51 ± 0.30	66.62±	22.05 ± 0.14	22.17 ± 0.02	22.11 ± 0.04	11.23 ± 0.06	11.33 ± 0.16	11.28 ± 0.18	3.03 ± 0.04	3.00 ± 0.05	3.01 ± 0.05
	LA	90.80 ± 0.30	95.12 ± 1.64	92.96 ± 0.60	67.27 ± 0.17	66.04 ± 0.25	66.66±	21.69 ± 0.17	22.73 ± 0.02	22.21 ± 0.02	11.04 ± 0.03	11.24 ± 0.18	11.14 ± 0.05	3.10 ± 0.05	2.91 ± 0.02	3.00 ± 0.04
*p*-value		0.82	0.13	0.97	0.97	0.49	0.43	0.62	0.74	0.94	0.16	0.83	0.19	0.65	0.57	0.93

SEM = standard error of the mean, interaction = source of starch × modified methods, CSC = cassava chip, WBT = winged bean tube, NaOH = Sodium hydroxide treated, CaOH_2_ = Calcium hydroxide treated, LA = lactic acid treated, C2 = acetic acid, C3 = propionic acid, C4 = butyric acid.

## Data Availability

The data that support the findings of this study are available from the corresponding author upon reasonable request.

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
