# Peer review of "Comparison of Cassava Chips and Winged Bean Tubers with Various Starch Modifications on Chemical Composition, the Kinetics of Gas, Ruminal Degradation, and Ruminal Fermentation Characteristics Using an In Situ Nylon Bag and an In Vitro Gas Production Technique"

_animals, 2023, doi:10.3390/ani13101640_

Round 1
Reviewer 1 Report (New Reviewer)
Line 18: please replace ‘immediate’ with ‘higher’
Line 19: replace ‘of more’ with ‘that is more’
Line 19-20: rewrite the sentence ‘However, many problems remain, particularly in cassava field crops, resulting in productivity fluctuations and price inconsistency’ the new sentence could be ‘However, inconsistency in production and fluctuations in prices encouraging animal nutritionist to search alternative of CSC’
Lien 23-25: please correlate these two sentences ‘However, the starch content may degrade rapidly in the rumen, thereby decreasing ruminal pH. It was discovered that steamed WBT might be useful to improve feed efficiency, which can decrease degradability and maintain rumen pH’ joining of these two sentences with some modification may improve the sentence structure and clear the meanings
Line 26: please provide full term of CSC and WBT on first seen in this section because this section seems independent in HTML
Line 29: remove the abbreviation CRD for completely randomized design
Line 30: are you not confusing starch with energy? ‘two energy sources’
Line 28-32: experimental treatments could be rewritten in better way ‘Ten treatments were formed according to a 2 × 5 factorial design in a completely randomized design (CRD). The first factor was the two energy sources: CSC and WBT. The second factor was based on five treatment methods: untreated, steam treatment, sodium hydroxide (NaOH) treatment, calcium hydroxide (CaOH2)treatment, and lactic acid (LA) treatment’ for example ‘ Experimental treatments were arranged in 2 × 5 factorial design with two sources of starch and five level of modification treatments. Two sources of starch were CSC and WBT, while, five modification treatments of starch were no modification treatment, steam treatment, sodium hydroxide (NaOH) treatment, calcium hydroxide (CaOH2)treatment, and lactic acid (LA) treatment’
Line 33: what do you mean by ‘ash content by…’ please correct it
Line 33: what do you mean by ‘but the CP content….’ Do you mean crude protein? If yes, please provide full term on first seen
Line 32-34: these results are not representing the exact results. For example, ‘The starch modification methods with NaOH and CaOH2 increased the ash content by, (P< 0.05)’ these results were for both starch sources? Please clarify
Line 38: remove ‘IVDMD’ because its not being used in this section again
Line 40: please remove ‘(NH3-N)’ because its not being used in this section again
Line 40-42: conclusion is not conclusive ‘In conclusion, compared to the CSC group, treating WBT with steam might be a more effective strategy for enhancing feed efficiency, decreasing degradability, and maintaining rumen pH’ I am unable to find the results of feed efficiency and pH in results of abstract
Line 51-53: please rewrite ‘However, many problems persist, particularly in cassava field crops, which produce productivity variations [3]. One of these variables is the impact of climate change on agricultural output, which causes price inconsistency and volatility’ meanings are not so clear
Line 56-57: please provide reference ‘it has been recommended that Psophocarpus tetragonolobus (winged bean (WB)) be used more widely, particularly in tropical areas’
Line 57: please don’t start sentence from abbreviation ‘WB has green pod productivity ….’
Line 58: crude protein has already been used on line 49, please abbreviate crude protein at line 49 ‘They have a high soluble carbohydrate content (75 to 85%) but a low crude protein content…..’
Line 61-62: please provide the reference ‘Usually, high proportions of carbohydrate sources are often used for high-producing ruminants to support high milk production and quick weight gain’ you may read the recent articles regarding this information
Line 61-65: one of your objective is to check rumen microbial population by your treatments but in this section that is missing. It should be mentioned during hypothesis construction of your study
‘Usually, high proportions of carbohydrate sources are often used for high-producing 61 ruminants to support high milk production and quick weight gain. However, the rapid fermentation of high-carbohydrate feeds containing high starch causes a drop in ruminal pH and increases the risk of rumen acidosis [8]. Therefore, much research has been done to find a suitable way of modulating the degradability of starch sources in the rumen’
You may read the recent article for example
Chen, D., Q. Tang, H. Su, H. Zheng, K. Chen and G. Zhong. 2021. Rumen microbial community and functions of rumen bacteria under different feeding regime. Pakistan Veterinary Journal 41: 341-346.
Line 88: please don’t start the sentence with abbreviation ‘CSCs were obtained from Khon Kaen….’
Line 89-90: its not clear ‘plant breeding researchers….’ You should properly mention the place from where WBT was procured
Line 91-92: this sentence is not clear ‘The steam treatment method was modified according to Srakaew et al. [8]’ please make it clear. Either you developed new method? Or you used the previous technique with some modification
Line 94-95: please see above comment for this statement too ‘The NaOH treatment method was modified from the 94 process of Srakaew et al. [8].’
Same as above comment ‘The CaOH2 treatment 96 method was modified from the procedure of Wanapat et al. [15]’
Line 104: is it dietary treatment or experimental treatment ‘Dietary treatments and experimental design’
Line 111: replace ‘dietary sample’ with ‘suitable word’
Line 116: Line 58: crude protein has already been used on line 49, please abbreviate crude protein at line 49 ‘They have a high soluble carbohydrate content (75 to 85%) but a low crude protein content…..’
Line 126: how could be ‘commercial concentrate (40%) and rice straw (60%)…’ diet is same with experimental treatment
Line 341: please don’t start the sentence with abbreviation ‘CSC has been used as an energy source in tropical ruminant rations….’
Line 419: please don’t start the sentence with abbreviation ‘WBTs have a high starch content, making them comparable…’
Author Response
Response to Reviewer 1
Line 18: please replace ‘immediate’ with ‘higher’
Response: Thank you for your suggestion. I appreciate you sharing it. We try our best to modify it already, please see the manuscript Line 18.
Line 19: replace ‘of more’ with ‘that is more’
Response: Thank you for your suggestion. I appreciate you sharing it. We try our best to modify it already, please see the manuscript Line 19.
Line 19-20: rewrite the sentence ‘However, many problems remain, particularly in cassava field crops, resulting in productivity fluctuations and price inconsistency’ the new sentence could be ‘However, inconsistency in production and fluctuations in prices encouraging animal nutritionist to search alternative of CSC’
Response: Thank you for your suggestion. I appreciate you sharing it. We modified it as your suggestion, please see the manuscript Line 19-20.
Lien 23-25: please correlate these two sentences ‘However, the starch content may degrade rapidly in the rumen, thereby decreasing ruminal pH. It was discovered that steamed WBT might be useful to improve feed efficiency, which can decrease degradability and maintain rumen pH’ joining of these two sentences with some modification may improve the sentence structure and clear the meanings
Response: Thank you for your suggestion. We have revised it as "Moreover, it was discovered that steamed WBT might be useful to improve feed efficiency, which could lower rapid starch degradability and maintain rumen pH." Please see in lines 23–24.
Line 26: please provide full term of CSC and WBT on first seen in this section because this section seems independent in HTML
Response: Thank you for your comment. We have modified it already, please see the manuscript.
Line 29: remove the abbreviation CRD for completely randomized design
Response: Thank you for your comment. We have revised it as per your suggestion.
Line 30: are you not confusing starch with energy? ‘two energy sources’
Response: Thank you for your suggestion. We try our best to modify it already, and we have tried to write and provide the first factor was the two energy sources: CSC and WBT. Please see the manuscript Line 30.
Line 28-32: experimental treatments could be rewritten in better way ‘Ten treatments were formed according to a 2 × 5 factorial design in a completely randomized design (CRD). The first factor was the two energy sources: CSC and WBT. The second factor was based on five treatment methods: untreated, steam treatment, sodium hydroxide (NaOH) treatment, calcium hydroxide (CaOH2) treatment, and lactic acid (LA) treatment’ for example ‘ Experimental treatments were arranged in 2 × 5 factorial design with two sources of starch and five level of modification treatments. Two sources of starch were CSC and WBT, while, five modification treatments of starch were no modification treatment, steam treatment, sodium hydroxide (NaOH) treatment, calcium hydroxide (CaOH2) treatment, and lactic acid (LA) treatment’
Response: Thank you for your suggestion. We have modified it as per your comment. Please see the manuscript Line 28-32.
Line 33: what do you mean by ‘ash content by…’ please correct it
Response: Thanks for your comment. We have modified it already, please see the manuscript Line 33.
Line 33: what do you mean by ‘but the CP content….’ Do you mean crude protein? If yes, please provide full term on first seen
Response: Thank you for your suggestion. We have revised it as per your comment. Please see the manuscript Line 33.
Line 32-34: these results are not representing the exact results. For example, ‘The starch modification methods with NaOH and CaOH2 increased the ash content by, (P< 0.05)’ these results were for both starch sources? Please clarify
Response: Thank you for your suggestion. We try our best to modify it already, and we have tried to rewrite the sentence as "The starch modification methods with NaOH and CaOH2 increased the ash content (P<0.05), whereas the crude protein (CP) content was lower after treatment with NaOH (P< 0.05)". Since in Table 1 was not shown an interaction effect for all parameters and those parameters (Ash and CP) were only affected by the starch modification methods. Thus, this sentence tried to mention only the impact of the starch modification methods on those parameters (Ash and CP). Please see the manuscript Line 32-34.
Line 38: remove ‘IVDMD’ because its not being used in this section again
Response: Thank you for your suggestion. We have modified it already, please see the manuscript Line 38.
Line 40: please remove ‘(NH3-N)’ because its not being used in this section again
Response: We appreciate your comment. We have modified it already, please see the manuscript Line 40.
Line 40-42: conclusion is not conclusive ‘In conclusion, compared to the CSC group, treating WBT with steam might be a more effective strategy for enhancing feed efficiency, decreasing degradability, and maintaining rumen pH’ I am unable to find the results of feed efficiency and pH in results of abstract
Response: Thank you for your comment. We have modified it to "In conclusion, compared to the CSC group and untreated treatment, treating WBT with steam might be a more effective strategy for enhancing feed efficiency by decreasing or retarding ruminal starch degradability and maintaining ruminal pH". We have considered several parameters for how steaming can improve the fermentation process. According to Table 2 and Table 5 as well as other tables, the effective degradability was lower in steamed WBT and the pH value was greater in the steaming method when compared to the untreated treatment, which might increase the chances of nutrient utilization in the lower gut and maintain ruminal pH, which also avoids rumen acidosis. Please see the manuscript Line 42-44.
Line 51-53: please rewrite ‘However, many problems persist, particularly in cassava field crops, which produce productivity variations [3]. One of these variables is the impact of climate change on agricultural output, which causes price inconsistency and volatility’ meanings are not so clear
Response: Thanks for your suggestion. We have modified to “Although cassava is widely used as a chewable feed in the tropics, there are still some problems with productivity variations from the effects of climate change on agricultural production, which also lead to instability in prices”. Please see the manuscript Line 51-53.
Line 56-57: please provide reference ‘it has been recommended that Psophocarpus tetragonolobus (winged bean (WB)) be used more widely, particularly in tropical areas’
Response: We appreciate your comment. We have modified it already, please see the manuscript Line 56-57.
Line 57: please don’t start sentence from abbreviation ‘WB has green pod productivity ….’
Response: Thanks for your suggestion. We have revised it already, please see the manuscript Line 57.
Line 58: crude protein has already been used on line 49, please abbreviate crude protein at line 49 ‘They have a high soluble carbohydrate content (75 to 85%) but a low crude protein content…..’
Response: Response: We appreciate your suggestion. We have modified it already, please see the manuscript Line 49 and 58.
Line 61-62: please provide the reference ‘Usually, high proportions of carbohydrate sources are often used for high-producing ruminants to support high milk production and quick weight gain’ you may read the recent articles regarding this information
Response: Thank you for your suggestion. We have modified it already, please see the manuscript Line 61-62.
Line 61-65: one of your objective is to check rumen microbial population by your treatments but in this section that is missing. It should be mentioned during hypothesis construction of your study
Response: We appreciate you sharing it. We provided the main objective was to evaluate the chemical composition, ruminal degradation, gas kinetics, and ruminal fermentation of CSC and WBT with various modified starches using an in situ degradability and in vitro gas production technique. Please see the manuscript Line 85-90.
‘Usually, high proportions of carbohydrate sources are often used for high-producing 61 ruminants to support high milk production and quick weight gain. However, the rapid fermentation of high-carbohydrate feeds containing high starch causes a drop in ruminal pH and increases the risk of rumen acidosis [8]. Therefore, much research has been done to find a suitable way of modulating the degradability of starch sources in the rumen’
You may read the recent article for example
Chen, D., Q. Tang, H. Su, H. Zheng, K. Chen and G. Zhong. 2021. Rumen microbial community and functions of rumen bacteria under different feeding regime. Pakistan Veterinary Journal 41: 341-346.
Response: We appreciate you sharing it. We try our best to modify it already, please see the manuscript Line 85-90.
Line 88: please don’t start the sentence with abbreviation ‘CSCs were obtained from Khon Kaen….’
Response: Thanks for your comment. We have modified it already, please see the manuscript Line 88.
Line 89-90: its not clear ‘plant breeding researchers….’ You should properly mention the place from where WBT was procured
Response: We appreciate your suggestion. We have modified it already, please see the manuscript Line 100-101.
Line 91-92: this sentence is not clear ‘The steam treatment method was modified according to Srakaew et al. [8]’ please make it clear. Either you developed new method? Or you used the previous technique with some modification
Response: Thanks for the comment. We used the previous technique of Srakaew et al. [8]. Please see the manuscript Line 91-92.
Line 94-95: please see above comment for this statement too ‘The NaOH treatment method was modified from the 94 process of Srakaew et al. [8].’
Response: Thank you for the comment. We used the previous procedure of Srakaew et al. [8].
Same as above comment ‘The CaOH2 treatment 96 method was modified from the procedure of Wanapat et al. [15]’
Response: Thanks for the comment. We used the previous procedure of Wanapat et al. [15]’
Line 104: is it dietary treatment or experimental treatment ‘Dietary treatments and experimental design’
Response: Thank you for your comment. We have modified it to "Experimental design and dietary treatments". We try to mention the experimental design first, then we mention the dietary treatments used in this study and their preparation, as well as the chemical composition analysis of the experimental diet.
Line 111: replace ‘dietary sample’ with ‘suitable word’
Response: Thank you for your suggestion. We have modified it already, please see the manuscript Line 111.
Line 116: Line 58: crude protein has already been used on line 49, please abbreviate crude protein at line 49 ‘They have a high soluble carbohydrate content (75 to 85%) but a low crude protein content…..’
Response: We appreciate your comment. We have modified it already, please see the manuscript Line 116.
Line 126: how could be ‘commercial concentrate (40%) and rice straw (60%)…’ diet is same with experimental treatment
Response: Thanks for the comment. In order to make it clear we modified to " All cattle received a concentrate diet at 1.0%BW (16.0% CP and 10.46 MJ of ME/kg DM) and rice straw twice daily ad libitum". Please seen in manuscript. The concentrate diet is differed to the experimental treatment, since we fed animal with basal diet as animal consumed before.
Line 341: please don’t start the sentence with abbreviation ‘CSC has been used as an energy source in tropical ruminant rations….’
Response: Thank you for your suggestion. We have modified it already, please see the manuscript Line 341.
Line 419: please don’t start the sentence with abbreviation ‘WBTs have a high starch content, making them comparable…’
Response: We appreciate your suggestion. We have modified it already, please see the manuscript Line 419.
Thanks so much for the valuable comments!

Reviewer 2 Report (New Reviewer)
Authors evaluated the cassava chips and winged bean tubers using different technological methods for starch modification on the ruminal degradation, fermentation, digestibility and gas kinetics using in vitro and in situ techniques.
The manuscript is well written and results are correctly supported by a proper discussion.
In general, the standard error or standard deviation should be provided in tables, figures and within the text. There is an overlapping table in the results chapter (Table 5 and Table 6).
I have some comments in order to improve the quality of the manuscript for reconsidering it for the publication. Find my suggestions in the attached PDF file.

Author Response
Response to Reviewer 2
Authors evaluated the cassava chips and winged bean tubers using different technological methods for starch modification on the ruminal degradation, fermentation, digestibility and gas kinetics using in vitro and in situ techniques.
The manuscript is well written and results are correctly supported by a proper discussion.
Response: We appreciate your kind words and comments. We have done our best to revise them in accordance with your comments and suggestions. Please see the manuscript, point-by-point response. Thanks again.
In general, the standard error or standard deviation should be provided in tables, figures and within the text. There is an overlapping table in the results chapter (Table 5 and Table 6).
Response: Thank you for your suggestions and we have all agreed with you. Present revised version we have added standard deviation into the Tables and also added error bar into Figure 1. In addition, Table 5 and 6 were modified already.
I have some comments in order to improve the quality of the manuscript for reconsidering it for the publication. Find my suggestions in the attached PDF file.
Response: We appreciate your kind words and comments. We have tried our best to following revised as comment in pdf file. Please see the manuscript, point-by-point response. Thanks again.
L19: Which is higher than.
Response: Thanks for the comment. We have modified it already, please see the manuscript Line 19.
L24: This description could be improved by underlining the most relevant results obtained from your study presented without technical terms.
Response: Thank you for your suggestion. We have modified it already, please see the manuscript Line 24.
L33: Something is missing in this sentence, please revise.
Response: We appreciate for your concern. We have modified it already, please see the manuscript Line 33.
L34-35: Was the CP content lower in both energy sources?
Response: Thanks for the comment. Yes, it was lower for both energy sources. According to Table 1, there is no interaction effect, while the single factor of modification treatments shows that the NaOH method resulted in a decrease in CP value. Thus, it means the CP content is lower in both energy sources. We have modified it already, please see the manuscript Line 34-35.
L49: Can you provide different references for these statements? Since 1-3 have already cited in the previous sentence.
Response: Thank you for your comment. We have revised it already, please see the manuscript Line 49.
L52-55: Can you provide another reference?
Response: Thanks for your comment. We modified it already, please see the manuscript Line 52-55.
L56: Avoid double brackets.
Response: We appreciate for your concern. We have modified it already, please see the manuscript Line 57.
L69: What is meant for "improving" digestion of resistant starch in the rumen?
Response: Thank you for suggestion. We try our best to modify as “By increasing resistant starch in the rumen, feed processing techniques including physical and chemical approaches can alter starch, reduce rumen degradable starch, and improve starch flow to the duodenum.” Please see the manuscript Line 86-88.
L74: The novelty of the study should be clarified.
Response: Thank you for suggestion. We have described more as “Although BWT contains a lot of starch, it can rapidly break down in the rumen, resulting in inadequate utilization. The novelty of the current work is due to the lack of knowledge used for controlling starch digestion in the rumen for BWT and appropriate tests for in vitro investigation. This study tested the hypothesis that a modified starch product obtained from WBT could be used as a potential energy source, decrease the rapid ruminal degradation rate, and maintain the ruminal pH. The aim was to evaluate the chemical composition, ruminal degradation, gas kinetics, and ruminal fermentation of CSC and WBT with various modified starches using an in situ degradability and in vitro gas production technique. Please see the manuscript Line 100-109”
L103: Which size?
Response: Thanks for the comment. We have modified it already, please see the manuscript Line 102-103.
L118: Are you sure that NFC refers to Non-neutral detergent carbohydrate?
Response: Thank you for suggestion. We have revised it to “Nonfibre carbohydrate” already. Please see the manuscript Line 118.
L124: Male or female?
Response: Thank you for suggestion. We have modified it already, please see the manuscript Line 124.
L141: Please describe how blanks have been prepared.
Response: Thank you for suggestion, we try our best to modify it already. These blank samples are those at 0 h of incubation and underwent a similar procedure for washing and drying in an oven.
L155: Is this formula according to Orskov and McDonald too?
Response: Thank you for the comment. we try our best to modify it already. Yes, in the same reference equation.
L159: Did you use different animals?
Response: Thanks for the comment. We use different experimental animal.
L172: Can you report the following method briefly?
Response: Thank you for the comment. We have added the protocol for method briefly, please see the manuscript Line 172-176.
L232: Avoid for listing non-significant p-values. Check it in all the manuscript.
Response: Thank you for your concern. We have modified it already, please see the manuscript Line 232.
L241: Table 1: Standard error or standard deviation should be provided for each mean. In addition, p-values should be listed at the bottom of the table.
Response: Thank you for your suggestion. Based on your suggestions, we have added a standard deviation value of each treatment. Please see Table 1-6.
L255-256: Please provide an error range.
Response: Thank you for suggestion. We have modified it already, please see the manuscript Line 260-261.
L262: Table 2: Please, provide p-values.
Response: Thanks for your suggestion. We have modified it already, please see the manuscript Line 260-261.
L268: Figure 1: Add the error ranges. Did you detect negative values?
Response: Thank you for suggestion, we try our best to modify it already, please see the Figure 1. Error bar has been added as you and Editor suggestion. In addition, the negative values are obtained when raw data was fitted to the model [19] for Y = a + b (1 – e(-ct))
L270-276: Figure 1. The figure should be listed after its citation in the text.
Response: Thank you for suggestion, we try our best to modify it already, please see the manuscript Line 270-276.
L290: Table 3: Provide p-values
Response: Thank you for the suggestion. P-value have been added for all Tables, please see in the manuscript.
L322: Provide always the standard error or deviation for each listed mean
Response: Thank you for suggestion. We try our best to modify it already, please see the manuscript Line 328-329.
L326: Table 5-6: Please revise these two tables that are overlapping.
Response: Thank you for suggestion, We have revised it as your suggestion, please see the manuscript in Table 5-6.
L347: Are in literature only two studeis that evaluated something similar?
Response: Thank you for suggestion, we try our best to modify it already. We discovered similar studies in the following references. Please see the manuscript Line 352.
L364-365: Was the contribution of minerals in these two solutions so relevant?
Response: Thank you for the comment. There is no relationship because of in this work evaluates two factors; a 2×5 factorial experiment design was carried out and set up using a completely randomized design (CRD). The interaction between the two compounds was not found. Diets used in the experiments had two starch energy sources, CSC and WBT, with five modified starch treatment methods: untreated, steam-treated, NaOH-treated, CaOH2-treated, and LA-treated. It was a separate study between the treatments of the individual compounds. Therefore, the contribution of minerals in these two solutions is not relevant.
L370: Clarify how it is meant the mentioned improvement.
Response: Thank you for suggestion. We try our best to clarify it. “It is possible that the processing procedure improves cereal starch by changes in structure and properties such as granular swelling, crystallite melting, birefringence loss, amylose leaching, and increased amylopectin concentration (Pu et ai., 2021)”.
L424-425: Could you clarify how can you measure a negative emission?
Response: The present results demonstrated that the intercept value of (a) was negative in this study. This was a result of the delay in ruminal microbial growth of the substrates during the early stage of incubation. The data show that there is a lag period after the soluble part of the substrate is ingested but before the cell walls are fermented [Chanthakhoun et al., 2012; Soriano et al., 2014]. Several researchers [Khazaal et al., 1993; Blummel and Becker, 1997; Suriyapha et al., 2021] have also stated that, when using mathematical models to match the kinetics of gas output, there were negative values for different substrates. It is understood that it is possible to use the absolute value of a, (|a|), to define the ideal fermentation of the soluble fraction.
L483: Try to rephrase this sentence to avoid repetitions.
Response: Thank you for suggestion. We have modified it already. Please see the manuscript Line 519-520.
L517-519: Why did you evaluate protozoa?
Response: Thank you very much. Previous studies (Eadie and Mann, 1970; Mackie et al., 1978) have suggested that protozoa play a beneficial role in high-concentrate diets stabilizing ruminal pH and reducing the rate of starch fermentation. Evaluating the role of ruminal protozoa in high-concentrate diets is important to elucidate the effects on ruminal and intestinal starch digestion. Thus, we study the parameter of protozoa count for modified starches on ruminal degradation and stabilizing ruminal pH.
L544-547: Based on your expertise, could it be convenient to include steamed WBT compared to CSC? Could it become a convenient alternative to CSC?
Response: Thank you very much. The advantages of including steamed make easy to recycle, multi-functionality, low production cost, starch can be easily modified and faster, and non-toxicity for animals (Deckardt et al., 2013). Moreover, Steam processing improved starch digestion in the rumen (Shen et al., 2015; Gómez et al., 2016), while treatment was reported to increase starch flow to the duodenum. It was discovered that steaming and alkaline treatment of cassava starches reduced ruminal degradability and enhanced by-pass starch, whereas non-processed maize resulted in by-pass starch being reduced. (Fadel-Elseed et al., 2003).

Reviewer 3 Report (New Reviewer)
This study by Unnawong et al. evaluated the impact of CSC and WBT with various starch modification methods on the chemical composition, ruminal degradation, gas production, in vitro degradability, and ruminal fermentation of feed using an in situ and in vitro gas production technique. The study is focused, well-designed and experimental procedures are clearly described.
The manuscript is clearly written, the discussion is conducted correctly, the conclusions are concise and reflect the results well, and I think the article is almost ready for publication (with minimal corrections, of course).
I consider the topic taken up to be important, especially taking into account the declining fodder base for ruminants caused by climate change. Research on the modification of starch into resistant starch is also important from the point of view of human nutrition - more and more people suffer from insulin resistance, diabetes or metabolic diseases associated with obesity. In their case, it is also important to provide the body with complex or difficult to break down carbohydrates.
The authors did not avoid some errors that are normal in large works. Please find my comments below.
Line 33 – “increased the ash content by, (P<0.05),” are you sure this sentence is spelled correctly? Shouldn't you give a specific figure by how much the ash level increased? Or remove the word 'by'?
Line 50 – ‘Cassava stars have a solubility’ - what solubility? High? Short?
Figure 1 – is unreadable. You should modify it, e.g. by increasing the intervals on the gas production scale or use a different chart format
Tables are a bit unreadable, please make spaces between each CSC and WTB and between interactions or separate them with a line. In addition, set the tables so that they are not divided by a new page, they will then be more readable
Table 6 - what is the 'mini table' below the table? Is it still part of the table? If so, what values does it refer to? I why is it in a different format and size than the rest?
Make sure that all Latin names used in the work are in italics.
Author Response
Response to Reviewer 3
This study by Unnawong et al. evaluated the impact of CSC and WBT with various starch modification methods on the chemical composition, ruminal degradation, gas production, in vitro degradability, and ruminal fermentation of feed using an in situ and in vitro gas production technique. The study is focused, well-designed and experimental procedures are clearly described.
The manuscript is clearly written, the discussion is conducted correctly, the conclusions are concise and reflect the results well, and I think the article is almost ready for publication (with minimal corrections, of course).
I consider the topic taken up to be important, especially taking into account the declining fodder base for ruminants caused by climate change. Research on the modification of starch into resistant starch is also important from the point of view of human nutrition - more and more people suffer from insulin resistance, diabetes or metabolic diseases associated with obesity. In their case, it is also important to provide the body with complex or difficult to break down carbohydrates.
The authors did not avoid some errors that are normal in large works. Please find my comments below.
Response: We would like to sincerely thanks Reviewer 3, who provided a positive comment and useful suggestion to improve this manuscript. In the present revised version, we have tried our best to modify as comments and suggestions. Please see the manuscript.
Line 33 – “increased the ash content by, (P<0.05),” are you sure this sentence is spelled correctly? Shouldn't you give a specific figure by how much the ash level increased? Or remove the word 'by'?
Response: We appreciate for your concern. We modified it already, please see the manuscript Line 33.
Line 50 – ‘Cassava stars have a solubility’ - what solubility? High? Short?
Response: Thanks for the comment. We modified to “Cassava starch has a high solubility and an immediate degradability rate in the rumen of more than 90%”. Please see the manuscript Line 50.
Figure 1 – is unreadable. You should modify it, e.g. by increasing the intervals on the gas production scale or use a different chart format
Response: Thank you for your comment and suggest changing to another format. However, the present figure would like to show trend of commutative gas production change at different hours and treatment. In case we would like to see details for the kinetic gas, it can be observed in Table 2.
Tables are a bit unreadable, please make spaces between each CSC and WTB and between interactions or separate them with a line. In addition, set the tables so that they are not divided by a new page, they will then be more readable
Response: Thanks for the comment. We try our best to modify it already, nevertheless, this is the official draft format from the journal that could be revised later.
Table 6 - what is the 'mini table' below the table? Is it still part of the table? If so, what values does it refer to? I why is it in a different format and size than the rest?
Response: We appreciate you sharing it. We modified it already, please see the manuscript in Table 6.
Make sure that all Latin names used in the work are in italics.
Response: Thank you for your suggestion. We modified it already, please see the manuscript.
Good luck!
Thanks so much for the valuable comments!

Round 2
Reviewer 1 Report (New Reviewer)
Thanks for revising the manuscript carefully
Author Response
Response to Reviewer 1
Thanks for revising the manuscript carefully.
Response: We appreciate the first reviewer for the great comments provided from the first round of revision and we hope that the present revised version will satisfy the rest reviewer.
Thanks so much for the valuable comments!

Reviewer 2 Report (New Reviewer)
I appreciated the authors' efforts that significantly improved the quality of the manuscript. I have other minor comments prior to reconsider the paper for the publication as listed below.
Line 59: It is not necessary to report again the abbreviation WB since it was already presented in the previous sentence (line 58).
Line 148: Can you specify that blanks are referred to samples at 0 h of incubation?
Line 166: Why did you need to use different animals for rumen liquor sampling?
Figure 1: I understand that negative values are referred to the calculation of selected model. I really appreciated your answer and my suggestion is to explain it along the results and discussion sections by providing references in order to avoid any misunderstood for the reader.
Table 6: this table should be amended since the font is too small and very difficult to read and interpret.
Round 1 - L517-519: Why did you evaluate protozoa?
Response: Thank you very much. Previous studies (Eadie and Mann, 1970; Mackie et al., 1978) have suggested that protozoa play a beneficial role in high-concentrate diets stabilizing ruminal pH and reducing the rate of starch fermentation. Evaluating the role of ruminal protozoa in high-concentrate diets is important to elucidate the effects on ruminal and intestinal starch digestion. Thus, we study the parameter of protozoa count for modified starches on ruminal degradation and stabilizing ruminal pH.
Round 2 - Thank you for your answer I suggest including this explanation in the text.
Round 1 - L544-547: Based on your expertise, could it be convenient to include steamed WBT compared to CSC? Could it become a convenient alternative to CSC?
Response: Thank you very much. The advantages of including steamed make easy to recycle, multi-functionality, low production cost, starch can be easily modified and faster, and non-toxicity for animals (Deckardt et al., 2013). Moreover, Steam processing improved starch digestion in the rumen (Shen et al., 2015; Gómez et al., 2016), while treatment was reported to increase starch flow to the duodenum. It was discovered that steaming and alkaline treatment of cassava starches reduced ruminal degradability and enhanced by-pass starch, whereas non-processed maize resulted in by-pass starch being reduced. (Fadel-Elseed et al., 2003).
Round 2 - Thank you for your answer, I suggest including this explanation in the text to provide a suggestion for further development of your findings.
Author Response
Response to Reviewer 2
I appreciated the authors' efforts that significantly improved the quality of the manuscript. I have other minor comments prior to reconsider the paper for the publication as listed below.
Response: We appreciate the Reviewer 2 for the great comments provided from the first round of revision and recent comment we have all agree with you and make a revise as your suggestion. Please see below our response.
Line 59: It is not necessary to report again the abbreviation WB since it was already presented in the previous sentence (line 58).
Response: We have modified from “The winged bean (WB)….” to “The WB…..” Please see in manuscript.
Line 148: Can you specify that blanks are referred to samples at 0 h of incubation?
Response: Thanks for your recommendation. We have specified in the text as “….a blank bag containing no sample for each removal time.” Please see in manuscript.
Line 166: Why did you need to use different animals for rumen liquor sampling?
Response: We want to collect rumen fluid from a larger number of representative animals. As a result, we revised the sentence to be easier to understand as, "In order to obtain a representative of rumen fluid, four native Thai beef cows were considered." Please see in manuscript.
Figure 1: I understand that negative values are referred to the calculation of selected model. I really appreciated your answer and my suggestion is to explain it along the results and discussion sections by providing references in order to avoid any misunderstood for the reader.
Response: Thanks for your recommendation. We have specified in the text as “The negative values were obtained after fitting the raw data to the model [19] for Y = a + b (1 – e(-ct)).” Please see in manuscript.
Table 6: this table should be amended since the font is too small and very difficult to read and interpret.
Response:
Round 1 - L517-519: Why did you evaluate protozoa?
Response: Thank you very much. Previous studies (Eadie and Mann, 1970; Mackie et al., 1978) have suggested that protozoa play a beneficial role in high-concentrate diets stabilizing ruminal pH and reducing the rate of starch fermentation. Evaluating the role of ruminal protozoa in high-concentrate diets is important to elucidate the effects on ruminal and intestinal starch digestion. Thus, we study the parameter of protozoa count for modified starches on ruminal degradation and stabilizing ruminal pH.
Round 2 - Thank you for your answer I suggest including this explanation in the text.
Response: Please accept my apologies for the incorrect response. Because the editor advised us to remove the protozoa data from the text (first round), the evaluation protozoa procedure was not required to be included in the text. Also, the protozoa results have been removed from the text and table.
Round 1 - L544-547: Based on your expertise, could it be convenient to include steamed WBT compared to CSC? Could it become a convenient alternative to CSC?
Response: Thank you very much. The advantages of including steamed make easy to recycle, multi-functionality, low production cost, starch can be easily modified and faster, and non-toxicity for animals (Deckardt et al., 2013). Moreover, Steam processing improved starch digestion in the rumen (Shen et al., 2015; Gómez et al., 2016), while treatment was reported to increase starch flow to the duodenum. It was discovered that steaming and alkaline treatment of cassava starches reduced ruminal degradability and enhanced by-pass starch, whereas non-processed maize resulted in by-pass starch being reduced. (Fadel-Elseed et al., 2003).
Round 2 - Thank you for your answer, I suggest including this explanation in the text to provide a suggestion for further development of your findings.
Response: Thank you very much for your recommendation. We have included some above information in the text as “In conclusion, considering WBT contains more starch and protein than CSC, steaming WBT might be a more successful strategy to enhance feed efficiency, decrease degradability, and maintain rumen pH, resulting in a convenient alternative to CSC.”
Thanks so much for the valuable comments!

This manuscript is a resubmission of an earlier submission. The following is a list of the peer review reports and author responses from that submission.
Round 1
Reviewer 1 Report
The research paper has many factors concerning however the authors should be improved the way how to present the data and try to show the Novel of the research study.
1. There are a lot of factors concerned in this research study however the presentation is not attractive to the audient. If the author rewrites the manuscript again and tries to show the strength of the research, it would be more attractive to the audience.
2. In the meanwhile, in the discussion part, the authors could not link each factor's results together. The author discussed the results by independent factors. Therefore the title of the publication is "Comparison ..... " so the author should explain in a short discussion again.
3. Regarding the result of the kinetic production of total gas, the author is able to present the data in the graph pattern instead of table data. It would be more interesting for the audience.
4. Line 332, would complete the sentence if the authors mentioned the number of the table again. Example "From the table 1 or Regarding the table 1".
Author Response
Response to Reviewer 1
The research paper has many factors concerning however the authors should be improved the way how to present the data and try to show the Novel of the research study.
Response: We appreciate your kind words, which serve as a strong endorsement of our work. We have done our best to revise them as your comments. Thanks again. Thanks again.
- There are a lot of factors concerned in this research study however the presentation is not attractive to the audient. If the author rewrites the manuscript again and tries to show the strength of the research, it would be more attractive to the audience.
Response: Thank you for your suggestions. The novelty of the work was provided in the section of Introduction and discussion. We try our best to modify it already, please see the manuscript.
- In the meanwhile, in the discussion part, the authors could not link each factor's results together. The author discussed the results by independent factors. Therefore the title of the publication is "Comparison ..... " so the author should explain in a short discussion again.
Response: Thanks for your suggestion. We have revised it as the comment. Some modifications have been discussed in the simple summary and discuss why WBR requires to compare to cassava. In the discussion part, we try to explain, and discussion link each factor's results together. Please see in manuscript.
- Regarding the result of the kinetic production of total gas, the author is able to present the data in the graph pattern instead of table data. It would be more interesting for the audience.
Response: We appreciate you sharing it. We have added a chart of the average value of cumulative gas (Figure 1), also mentioned and explained in the results section. Please see the manuscript.
- Line 332, would complete the sentence if the authors mentioned the number of the table again. Example "From the table 1 or Regarding the table 1".
Response: Thanks for your suggestion. We have revised it as the comment.
Thanks so much for the valuable comments!
Reviewer 2 Report
Dear authors
I have reviewed your manuscript. The theme is good and contributes to generate more knowledge about alternative feeds for livestock. In general, I consider that your experiment was conducted properly; however, there are some aspects that must be taken care of: 1. The use of the English language, 2. The writing, for which I recommend requesting the support of a style corrector and an English language editor, 3. The tables do not contain the correct information (so it is not possible to know if there was an effect of the interaction on the carbohydrate source and the processing), 4. It seems that there are calculation errors in some results presented.
Additionally, in the following lines, I mention some specific things.
Simply summary: Too short; there are no results or conclusions. I recommend going deeper
Abstact: very long. Include results.
Introduction
Line 48-51: The first paragraph is irrelevant and should be deleted.
Line 52: Indicate why the use of cassava chips is common in Thailand.
Line 60: Psophocarpus tetragonolobus is the scientific name, so it cannot be said "it is also known as". Correct.
Line 93-95: Indicate where the universities are located.
Materials and methods
Line 176: Indicate how many runs were made.
Line 186: Why did you analyze pH, ammoniacal N and protozoan count at 2 and 4 h of incubation?
Results
The tables need to be improved: Unify the size of the columns and make them wider. Use the P value to report the effect of starch source and processing.
Line 224: why is it claimed that steam increases the energy content of both carbohydrate sources? The results do not show it so. Correct.
Table 2 and 4. Explain if it is correct that the degradability of organic matter is greater than that of dry matter.
Table 6: Why does the sum of the values ​​of C2, C3 and C4 give 100 if they are supposed to be expressed in moles/100 moles? It should be taken into account that there are other AGVs, in addition to acetic, propionic and butyric. Apparently, the data is given in proportion. Correct.
Discussión and conclusions
OK
Author Response
Response to Reviewer 2
Dear authors
I have reviewed your manuscript. The theme is good and contributes to generate more knowledge about alternative feeds for livestock. In general, I consider that your experiment was conducted properly.
Response: We appreciate your kind words and comments. We have done our best to revise them in accordance with your comments and suggestions. Below is a detailed, point-by-point response. Thanks again.
However, there are some aspects that must be taken care of:
- The use of the English language,
Response: Thank you for your kind statements. We have been proofread English by a Native speaker which is provided by the American Manuscript Editor Company, located in Washington State, USA (Certificate Verification Key:295-723-438-051-971) and please see the revision of English in the Text.
- The writing, for which I recommend requesting the support of a style corrector and an English language editor,
Response: Thank you for your kind suggestion. We have done our best to revise them in accordance with your recommendation and suggestions. Thanks!
- The tables do not contain the correct information (so it is not possible to know if there was an effect of the interaction on the carbohydrate source and the processing),
Response: Thank you for suggesting very helpful advice for this experiment. We have already checked all of the information data. We confirmed the data's results are correct according to the experimental design. In this work evaluates two factors, a 2×5 factorial experiment design was carried out and set up using a completely randomized design (CRD). Diets used in the experiments had two starch energy sources, CSC and WBT, with five modified starch treatment methods: untreated, steam-treated, NaOH-treated, CaOH2-treated, and LA-treated. The results of the current study in the Table, where shown interaction = source of starch × modified methods, thus it is possible to know if there was an effect of the interaction on the carbohydrate source and the processing. Please consider again or if you have any more specific suggestions please let we know.
- It seems that there are calculation errors in some results presented.
Additionally, in the following lines, I mention some specific things.
Response: Thank you for your comment. We have done our best to revise them in accordance with your comments and suggestions. Below is a detailed, point-by-point response. Thanks!
Simply summary: Too short; there are no results or conclusions. I recommend going deeper
Response: Thanks, we have revised it as the comment.
Abstract: very long. Include results.
Response: Thanks, we have modified it as the comment. Please see in the abstract.
Introduction
Line 48-51: The first paragraph is irrelevant and should be deleted.
Response: Thanks, we have revised it as the comment.
Line 52: Indicate why the use of cassava chips is common in Thailand.
Response: Thank you for your great recommendation. We have revised it as the comment. Please see the introduction. Please see in line 58-61.
Line 60: Psophocarpus tetragonolobus is the scientific name, so it cannot be said "it is also known as". Correct.
Response: Thanks, we have revised it as the comment.
Line 93-95: Indicate where the universities are located.
Response: Thanks, we have revised it as the comment.
Materials and methods
Line 176: Indicate how many runs were made.
Response: Thanks, we have revised it as the comment. The present study was performed at different incubation intervals with two sub-experiments using a nylon bag measurement and gas production technique. In both sub-experiments, a 2×5 factorial experiment design was carried out and set up using a completely randomized design (CRD), with three replication runs. Please seen in line 151.
Line 186: Why did you analyze pH, ammoniacal N and protozoan count at 2 and 4 h of incubation?
Response: Thank you very much. We have analyzed pH, ammoniacal N, and protozoal count at 2 and 4 h of incubation because this study investigated the influence of starch that rapidly degrades, and those times try to simulate an optimal feed digestion time and an optimal dynamic time for ruminal microbe growth like the cattle's rumen. In addition, it is a universal and reliable method. A previous study has analyzed pH, ammoniacal N, and protozoan count at 2 and 4 h of incubation. Suriyapha et al. (2021; https://doi.org/10.3390/fermentation7030120); Chanjula et al. (2021; https://doi.org/10.3390/fermentation7040218).
Results
The tables need to be improved: Unify the size of the columns and make them wider. Use the P value to report the effect of starch source and processing.
Response: Thanks, we have revised it as the comment.
Line 224: why is it claimed that steam increases the energy content of both carbohydrate sources? The results do not show it so. Correct.
Response: From the table, it can be seen that the GE increased when treated with steam due to the sample's rich starch gelatinization and amylopectin concentration. This result was similar to Pu et al.’s [32] reports, who discovered that with the increased degree of gelatinization in corn starch. It is possible that the processing procedure improves cereal starch. Please see in line 393-397.
Table 2 and 4. Explain if it is correct that the degradability of organic matter is greater than that of dry matter.
Response: Thank you very much. Usually, the degradability of organic matter is greater than the degradability of dry matter which were also noted by Suntara et al. (2022; https://doi.org/10.3390/fermentation8050209). The dry matter is defined as the weight loss of samples when dried in an oven at above 100∘C for 12-24 hours. Organic matter defines as the weight loss of dry matter when combustion (dry matter minus ash content). Dry matter consisted of all nutrients, whereas organic matter consisted of all nutrients except ash (Anam Al-Arif et al., 2017).
Table 6: Why does the sum of the values ​​of C2, C3 and C4 give 100 if they are supposed to be expressed in moles/100 moles? It should be taken into account that there are other AGVs, in addition to acetic, propionic and butyric. Apparently, the data is given in proportion. Correct.
Response: The sum of the values of C2, C3 and C4 give 100, they are presented as % of total VFA (acetic, propionic, and butyric). In the present study, we investigated only the main types of VFA (acetic, propionic, and butyric) but we did not investigate and report other VFA profiles (isobutyric, isovaleric, and valeric). Moreover, we analyzed VFA proportionally, so it was expressed in %, equivalent to 100, not in concentration.
Discussión and conclusions
OK
Response: Thank you for your kind statements, which are a fantastic support of our work. Thanks again!
Thanks so much for the valuable comments!

Round 2
Reviewer 1 Report
Regarding the revised version, the authors significantly need to improve the manuscript eg. Data analysis, and the information to link each aspect. The explanation is still not clear for the relationship between individual factors after revising manuscript version 2. The authors proposed to show in the title that the modification of chemical composition has an impact on gas kinetic production, ruminal degradation, and fermentation. Unfortunately, the authors can not explain or link the key information of "modification of chemical composition" with other factors, as can be seen in the discussion part.
The authors tried to modify some data of SEM in many tables of version 2, to show the performance of the results by keeping the raw data the same as in version 1.
Many errors have been showing in the last version example of ...
figure 1. there are two line symbols of NaOH, CaOH2, and LA.
page 8. table 4. at SEM of IVDM at SEM >>> .12.15
table 6. the table is not fit on the page after being provided in the PDF format, there is some data missing.
Author Response
Response to Reviewer 1
Regarding the revised version, the authors significantly need to improve the manuscript eg. Data analysis, and the information to link each aspect. The explanation is still not clear for the relationship between individual factors after revising manuscript version 2. The authors proposed to show in the title that the modification of chemical composition has an impact on gas kinetic production, ruminal degradation, and fermentation. Unfortunately, the authors can not explain or link the key information of "modification of chemical composition" with other factors, as can be seen in the discussion part.
Response: Thank you for your suggestion and kindly recommend discussing more detail in order to link the key information of "modification of chemical composition" with other factors. However, the present work did not focus on the “modification of chemical composition”, but “comparison of various starch modification methods” has been elucidated. The title of this research is “Comparison of cassava chips and winged bean tubers with various starch modifications on chemical composition, the kinetics of gas, ruminal degradation, and ruminal fermentation characteristics using an in situ nylon bag and an in vitro gas production technique”. Our objective is to assess the impact of CSC and WBT with various starch modification methods on the chemical composition, ruminal degradation, gas production, in vitro degradability, and ruminal fermentation of feed using an in situ and in vitro gas production technique. Thus, discussion with various starch modification methods (untreated, steam treatment, sodium hydroxide (NaOH) treatment, calcium hydroxide (CaOH2) treatment, and lactic acid (LA) treatment) on chemical composition, ruminal degradation, gas production, degradability, and ruminal fermentation have been discussed in the present research. As you can observe in the discussion section, we tried to discuss how the biological mechanism of each starch modification method influences parameter measurements. For example starch modification methods on CP content: L344-349: “In this study, the CP content was significantly different based on the modification methods with NaOH treatment, which has negative side effects and results in a lower CP value (Table 1). This could be due to the starch granules swelling after being exposed to NaOH, resulting in lower CP content essential amino acids (cystine and lysine), amylose, or amylopectin, which creates a continuous gelled matrix that encircle the fragments of starch granules that rupture and leach out of the structure [28,29].” Etc.
However, in order to make it clearer, we have provided more explanation in the section of the discussion, please see the manuscript with track change mode.
The authors tried to modify some data of SEM in many tables of version 2, to show the performance of the results by keeping the raw data the same as in version 1.
Response: Thanks for your suggestion. The SEM and P-value were changed as a result of an editor's suggestion in Round 1 to use another procedure to compare treatment means, such as the Tukey test or the Least Significant Difference. As a result, we re-analyzed the data to compare treatment means using the Tukey test and discovered that some of the SEM values or p-values had changed. As a result, the revised version has been altered as shown in the manuscript.
Many errors have been showing in the last version example of ...
figure 1. there are two line symbols of NaOH, CaOH2, and LA.
Response: I appreciate you sharing it. We tried our best to modify it as comments, please see Figure 1.
page 8. table 4. at SEM of IVDM at SEM >>> .12.15
Response: Thanks for your suggestion. We have revised it as per the comment.
table 6. the table is not fit on the page after being provided in the PDF format, there is some data missing.
Response: Thanks for your suggestion. We have tried to reduce the size of the text in Table 6 to make some data not missing. Please see the manuscript.
Thanks so much for the valuable comments!

Reviewer 2 Report
Dear Authors,
Thank you for your response. Everything is ok, except the following:
- Tables: same column size
- Figure 1: Correctly Identify treatments within the figure (for example: CSC raw, CSC NaOH, CSC CaOH2, etc). Delete decimals. Include as footnote the following: CSC= cassava chip, WBT = winged bean tuber, Raw = raw stuff (untreated), NaOH = Sodium hydroxide treated, CaOH2 = Calcium hydroxidetreated, LA = LA = lactic acid treated.
Good luck!
Author Response
Response to Reviewer 2
Dear Authors,
Thank you for your response. Everything is ok, except the following:
Response: We appreciate your kind words and comments. We have done our best to revise them in accordance with your comments and suggestions. Below is a detailed, point-by-point response. Thanks again.
- Tables: same column size
Response: Thanks for your suggestion and we tried our best to modify it already. Please see the manuscript.
- Figure 1: Correctly Identify treatments within the figure (for example: CSC raw, CSC NaOH, CSC CaOH2, etc). Delete decimals. Include as footnote the following: CSC= cassava chip, WBT = winged bean tuber, Raw = raw stuff (untreated), NaOH = Sodium hydroxide treated, CaOH2 = Calcium hydroxide treated, LA = LA = lactic acid treated.
Response: I appreciate you sharing it. We tried our best to modify it already, please see Figure 1.
Good luck!
Response: Thank you for your best wishes and we hope that the present revised version can be considered for publication.
Thanks so much for the valuable comments!
